



# Transfer learning for landslide susceptibility modelling using domain adaptation and case-based reasoning

Zhihao Wang[1], Jason Goetz[1], Alexander Brenning[1]

[1] Friedrich Schiller University Jena, Department of Geography, Loebdergraben 32, 07743 Jena, Germany

*Correspondence to*: Zhihao Wang (zhihao.wang@uni-jena.de)

**Abstract.** Transferability of knowledge from well-investigated areas to a new study region is gaining importance in landslide hazard research. Considering the time-consuming compilation of landslide inventories as a prerequisite for landslide susceptibility mapping, model transferability can be key to making hazard-related information available to stakeholders in a timely manner. In this paper, we compare and combine two important transfer-learning strategies for landslide susceptibility

modelling: case-based reasoning (CBR) and domain adaptation (DA). CBR gathers knowledge from previous similar situations (source areas) and applies it to solve a new problem (target area). DA, which is widely used in computer vision, selects data from a source area that has a similar distribution to the target area. We assess the performances of single- and multiple-source CBR, DA and CBR-DA strategies to train and combine landslide susceptibility models using generalized additive models (GAMs) for 10 study areas with various resolutions (1 m, 10 m and 25 m) located in Austria, Ecuador, and Italy. The

performance evaluation shows that CBR and combined CBR-DA based on our proposed similarity criterion was able to achieve performances comparable to benchmark models trained in the target area itself. Particularly the CBR strategies yielded favourable results in both single- and multi-source strategies. DA tended to have overall lower performances than CBR; yet, it had promising results in scenarios when the source-target similarity was low. We recommend that future transfer learning research for landslide susceptibility modelling can build on the similarity criterion we used, as it successfully helped to achieve

landslide susceptibility model transfers by discovering suitable training datasets from various regions.



## 1 Introduction

Landslides are a not only common, but one of the most severe and critical natural hazards in mountain areas. Globally, the destruction caused by landslides continues to have severe effects on human activity and life (Froude and Petley, 2018; Haque et al., 2019). Landslide susceptibility mapping, the modelling of areas prone to landslide occurrence, is an effective method to
25 assist land managers in decision making aimed at minimizing landslide risk. One of the most challenging aspects of building data-driven landslide susceptibility models is establishing the landslide inventory data for model training and testing (Lin et al., 2021). Landslide inventories from different areas can provide informative knowledge of landslides – even in the case of older data, certain parts of the data can be still reused (Petschko et al., 2016). Additionally, previous landslide as well as ecological studies have pointed out that model transferability can aid in the prediction in adjacent regions (i.e., regional
susceptibility modelling) and have the potential to improve process understanding (Sequeira et al., 2016; Wenger and Olden, 2012; Rudy et al., 2016).

Machine learning is currently the most commonly applied method in research for solving the problem of landslide prediction (Goetz et al., 2015; Merghadi et al., 2020). Traditional machine learning operates on the condition that the training and test data are taken from the same input feature space and data distribution (Pan, 2014). In the case of spatial and temporal
predictions, this means that most fitted machine learning models are limited to the spatial and temporal bounds of the input data. Thus, when extrapolating or transferring traditional machine learning model to new spatial and temporal domains, model performance can be degraded due to differences in feature space and/or data distributions (Shimodaira, 2000; Pan and Yang, 2010; Yates et al., 2018).

A successful model transfer does not necessarily rely solely on the extent of geographic or temporal separation, but rather on
the similarity of the environmental conditions between the source and target areas (Yates et al 2018). The field of transfer learning offers various techniques to exploit this observation, which have yet to be fully utilized by the spatial modelling communities – including landslide susceptibility modelling. Transfer learning techniques such as domain adaptation (DA) and case-based reasoning (CBR) have been developed to select the best data and corresponding models from source areas for predicting in a spatially and or temporally distinct target area.

The general concept of transfer learning is to solve new problems by applying knowledge gained from previous experiences solving similar problems. That is, transfer learning has the potential to allow us to take existing knowledge of landslide occurrence from previous modelling experiences and apply it to new locations lacking any landslide data. Thus, this approach has incredible potential to minimize the considerable time and effort needed for building landslide inventories for susceptibility modelling in new areas, especially in large and geographically remote areas landslide mapping and detection is particularly
challenging.



Landslide inventories from different source areas usually contain many different cases, each of which have a large amount of information. The problem is that not all cases are suitable for the new target task and processing the large amount of information for each case is of high cost in terms of time and effort. Therefore, it is desirable to compare the overall characteristics of each case to transfer the appropriate knowledge. CBR is a method to solve these problems by identifying similar cases and applying them to a new target area. This CBR similarity analysis can be performed by considering various attributes, such as data structure and topographic characteristics, which has been widely used in environmental sciences (Liang et al. 2021; Liang et al. 2020; Qin et al. 2016; Shi et al. 2004). In contrast, instead of finding best cases using the overall similarity of source areas to a target, which is done by CBR, DA transfer learning techniques can be applied to select the observations within a source area that match the data distribution of the target area. Previous applications of CBR in geosciences have focused on selecting one source area to transfer a target area (Qin et al 2016; Liang et al 2021). Yet, there is also potential for using CBR and DA to combine cases from multiple source areas to generate transferable models.

The objective of this study is to assess the potential of transfer learning using CBR and DA techniques for enhancing model transferability of machine-learning landslide susceptibility models. In particular, we evaluate the performance of transferred susceptibility models using DA, CBR and a combined CBR-DA technique, as well as the sensitivity of these methods to spatial resolution. We consider two scenarios for training landslide susceptibility models: only one source area available (single-source) and multiple source areas available for model training (multi-source). We examine both scenarios of these methods and compare them to benchmark situations, where susceptibility models are applied to a new target area without using transfer learning techniques.

## 2 Methods and Data

In transfer learning, the general goal is to train a model $f$ on data from one or multiple source areas $S = \{S_1, S_2, \ldots, S_N\}$ in order to apply it to make predictions in an unseen target area $T$ with $N_t$ observations regardless of spatial and temporal differences. A source area $S_i$ consists of $N_{S_i}$ observations of a set of predictors, $x_j$, and the corresponding labels $y_j$ (e.g., landslide or non-landslide), $j = 1, \ldots, N_{S_i}$.





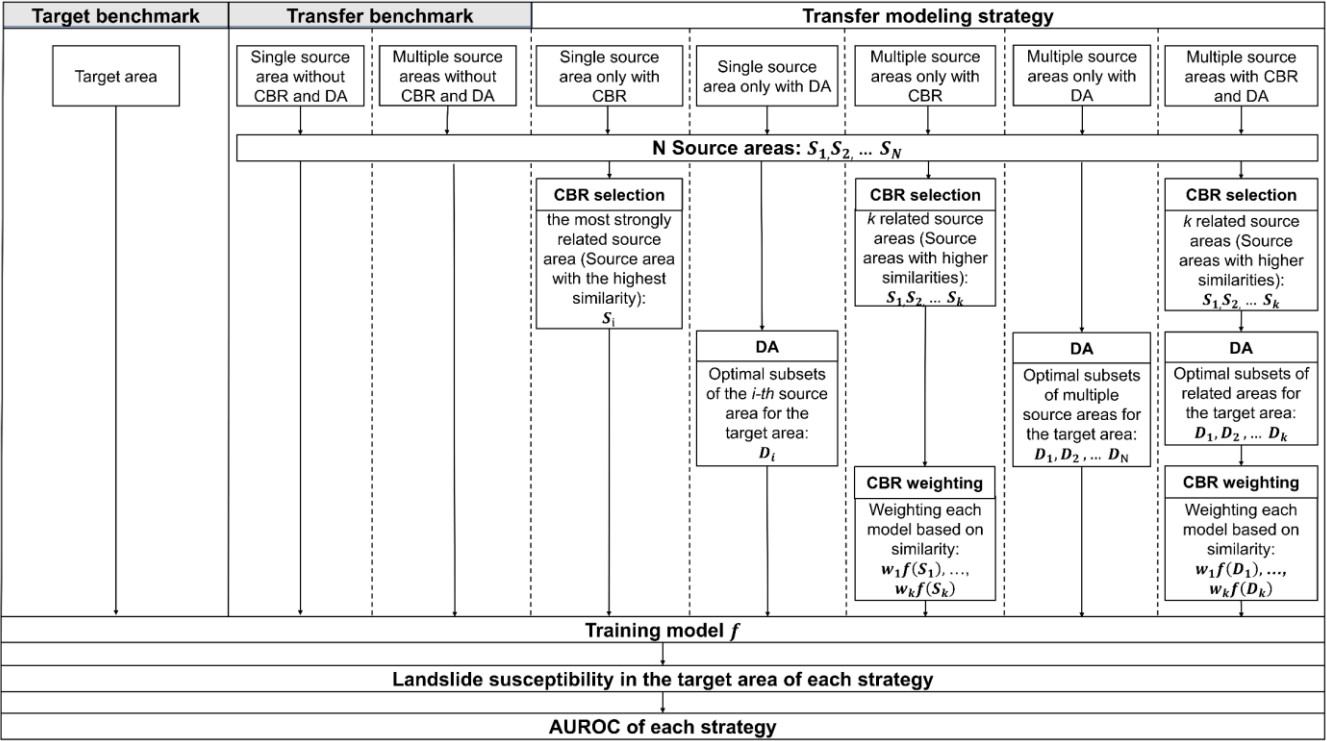

**Figure 1: Flow chart of transfer modelling strategies and benchmarks for landslide susceptibility mapping in a target area. Case-based reasoning (CBR) involves selection and weighting steps. In the single-source situation, weighting does not apply. Domain adaptation (DA) can be used by itself or combined with CBR to select source areas.**

Altogether, we evaluate five different transfer learning strategies for landslide susceptibility modelling that consider the use of data from single source area, multiple source areas, which are applied to CBR, DA, and both combined (CBR-DA) (Fig. 1).

To assess the relative performance of the transfer learning strategies, we include benchmark landslide susceptibility models that are simply trained using data from a single source area (single-source transfer benchmark), multiple source areas (multi-source transfer benchmark) and the target area (target benchmark), and then applied to the target area. In the case when multiple source areas were used, the benchmark transfer model was calculated by averaging the model predictions of multiple source areas without weighting (Table 2). The target benchmark, which is trained and tested with all of the target data, is meant to

represent an overoptimistic yet potentially obtainable performance for a given target area.

## 2.1 Case-based reasoning method

In machine learning, CBR is one of the most well-known methods for solving a new problem by referring to similar cases, which can translate the knowledge from geographical space to parameter space (Shi et al., 2004; Shi et al., 2009; Hammond, 2012). It finds cases in a data collection that are similar to the current case in terms of metadata and/or data distribution, and

then adopts those similar cases for training models (Liang et al., 2020a). This method has been reported to reduce the users'



modelling efforts while achieving good performances in use cases involving terrain attributes (Liang et al., 2020b; Qin et al., 2016).

CBR strategies are designed to find source areas $S = \{S_1, S_2, ..., S_k\}$ that are most similar to the target area based on statistical summary information and metadata; these selected areas are referred to as *related* areas. In generating a CBR model, the individual models trained on the selected source areas are combined as a weighted sum

$$f(\mathbf{x}) = \sum_{i=1}^{k} w_i f_i(\mathbf{x}) \tag{1}$$

whose weights $w_i$ correspond to similarity scores, normalized to sum up to 1. The individual models $f_i$ may be trained using conventional sampling strategies as well as DA strategies, both of which are described below in detail.

Generally, CBR consists of the case problem and the corresponding case solution parts (Liang et al., 2020b; Qin et al., 2016). In our study, the problem part for formalizing the similarity of areas in landslide susceptibility modelling is to contrive a way to adequately describe the data and areas' contextual information, such as how a study area's spatial data can describe the pattern of landslide occurrence.

In applying CBR, it is first necessary to define and calculate similarity measures for relevant attributes that describe the data distributions of the source and target areas. In this study, we chose geological characteristics, spatial resolution and topographic characteristics. Similarities in each attribute were estimated based on similarity function (Table 1).

The geological characteristics of a region are an essential factor that influences multiple landslide conditioning factors such as the geomechanical and hydrological properties of hillslopes (Segoni et al., 2020). Considering the difficulties in matching geological descriptor such as heterogeneous chronostratigraphic units in different areas, we chose a simplified approach as a first-order approximation. Specifically, we used an indicator method that is based on whether the main rock types (igneous, sedimentary and volcanic rocks) coincide in source and target areas.

Topographic conditions were described by measures of total relief, standard deviation of slope angle, and mean slope angle (Wang et al., 2019). Total relief describes the overall terrain situation of a study area by subtracting the minimum elevation from the maximum elevation within the study area. The relief, which reflects the macroscopic characteristics of surface topography in a large area, has a good relation with landslide susceptibility (Wang et al., 2010). The standard deviation of slope is used to describe the topographic complexity of a study area. It is one of the most influential topographic variables in landslide susceptibility studies (e.g., Van Den Eeckhaut et al., 2012).



**Table 1 Similarity functions for the attributes used in CBR to identify related source areas: Geological characteristics, data characteristics, and topographic characteristics of study area**

| Factor group | Attribute | Similarity function | Description |
|---|---|---|---|
| Geological characteristics | Igneous Sedimentary Metamorphic | $Sim = \frac{1}{3}\sum_g I_g$ | where $I_g = 1$ if unit $g$ present or absent in the source *and* the target area, and 0 otherwise. |
| Data characteristics | Resolution [m] | $Sim = 2^{-(2\|log_{10} R_t - log_{10} R_s\|)^{0.5}}$ <br> or $Sim = 1$ | where the similarity is 1 if the resolution of source area is smaller than in the target area, otherwise $Sim$. $R$ is the DEM resolution. |
| Topographic characteristics | Total relief [m] | $Sim = 1 - \frac{\|Relief_t - Relief_S\|}{\max(8848 - Relief_t, Relief_t)}$ | where $Relief$ is the total relief. |
| | Standard deviation of slope | $Sim = 2^{-(2\|log_{10} Sd_t - log_{10} Sd_S\|)^{0.5}}$ | where $Sd$ is the standard deviation of slope angle. |
| | Mean slope [º] | $Sim = 1 - \frac{\|Slope_t - Slope_S\|}{\max(40° - Slope_t, Slope_t)}$ | where $Slope$ is the mean slope. |

**Note: Sim is the similarity of each individual attribute between the target area $t$ and a source area $S$, which is in [0,1]. The following constants were used for normalization: 8848 m is the elevation of Mount Everest. For mean slope, 40° can properly cover the mean slope in all study areas.**

The similarity values obtained for each factor were combined into a single indicator by taking their minimum (Qin et al., 2009; Qin et al., 2016; Zhu and Band, 1994). In this study, for a given target area we refer to source areas that have an overall (i.e. minimum) similarity score ≥ 0.65 as *related* source areas.

## 2.2 Domain Adaptation

The general machine-learning approach of domain adaptation (DA) aims to solve a learning problem in the target area by utilizing data from different source areas to construct a learning sample (Wang and Deng, 2018). A latent feature space is defined in which the source and target areas have the same distribution, and as a consequence, classifiers trained on labelled data from source areas are likely to perform well in the target area (Baktashmotlagh et al., 2013; Patel et al., 2015; Wilson and Cook, 2020). There are supervised DA techniques that require labelled data from the target area, and unsupervised methods that do not require such data (Ben-David et al., 2010; Courty et al., 2017). We adopt unsupervised DA in our study because its smaller data requirements seem more appealing for practical applications.

DA used in our study is a strategy for selecting instances $D_i \subset S_i$ (i.e., sample locations or grid cells for training) from a source area $S_i$ in such a way that their distribution is more similar to the target area's data distribution. In situations with multiple sources areas, DA is applied to each of them independently to obtain instance sets $D = \{D_1, D_2, ..., D_k\}$ on which k models are trained. The predictions from these models are either averaged (referred to as "plain" DA), or their weighted average is calculated when combined with source-area selection by CBR. DA is conventionally used as a single-source strategy, which is also included in this study for comparison even though multi-source strategies may seem more appealing in real-world applications.





Many DA strategies for transferring or weighting features can result in models that are difficult to interpret in terms of the geophysical process's modelled influence on the response. Also, not all instances from different source areas may be suitable for transferring to a target area (Gong et al., 2013; Jiang and Zhai, 2007; Long et al., 2013). Thus, the landmark-based domain adaptation (LBDA) approach (Gong et al., 2013) was applied in our study. This method selects the instances (or landmarks) from source areas with the same or similar distribution as the target area without creating new predictors. It aims at minimizing

the difference in sample means in latent feature space.

In our study, considering computational constraints, a randomly selected set of 50,000 unlabelled (landslide and non-landslide) points $x_n$ from the target area were used as reference points, and were compared to a randomly selected set of 30,000 labelled $(x_{S,m}, y_{S,m})$ (landslide and non-landslide) points from the source area $S$ as reference points to select a subset with similar data distribution as the target area. In the case of some of the smaller source areas in our study, all observations were used for subset

selection.

Domain adaptation selects training data by formally solving the following optimization problem

$$min \left\| \frac{1}{\sum_{m=1}^{N_S} \alpha_{S,m}} \sum_{m=1}^{N_S} \alpha_{S,m} \phi(x_{S,m}) - \frac{1}{N_t} \sum_{n=1}^{N_t} \phi(x_{t,n}) \right\|_H^2 \quad (2)$$

subject to $\frac{1}{\sum_{m=1}^{C} \alpha_{S,m}} \sum_{m=1}^{C} \alpha_{S,m} y_{S,m} = \frac{1}{N_S} \sum_{m=1}^{C} y_{S,m}$ \quad (3)

where indicator variables $\alpha = \{\alpha_m \in \{0,1\}, m = 1, \ldots, N_S\}$ are to judge whether landslide/non-landslide point in the source area is a landmark for minimizing the difference between source and target areas in the latent feature space. When $\alpha_m$ is 1, $(x_{S,m}, y_{S,m})$ is regarded as a landmark, that is, a landslide/non-landslide point that can provide valuable information for the

landslide susceptibility model of the target area. $\phi$ is a nonlinear feature function to map x to a Reproducing Kernel Hilbert Space (Gretton et al., 2006). Following Gong et al. (2013, 2017), Gaussian RBF kernels are used for $\phi$ in our study. And $C$ is the number of landslide or non-landslide points. The collection $\alpha$ is chosen when the eq. (1) is minimized, that is, the difference is minimized. Eq. (2) is the constraint that considers the distribution of labels in the selected landmarks. Then eq. (1) can be solved efficiently with convex optimization.

Finally, we will set a threshold for selecting landmarks. Ideally, the distribution of landslide/non-landslide points in the source and the target areas is the same, meaning that each landslide/non-landslide point in the source area can be a landmark. Therefore, we use $1/N_S$ as the criterion for obtaining the selected landmarks.



**Table 2 Transfer strategies and benchmarks adopted in our study.**

| Transfer Strategies | Final predictive model $f$ | Description |
|---|---|---|
| Single source area with DA (single-source DA) | $f_i(D_i)$ | The final prediction model is trained on the DA-derived subset of data from each source area. |
| Single source area with CBR (single-source CBR) | $f(S_{highest})$ | The final prediction model is trained using all data from the *most strongly related* source area. |
| Multiple source areas only with CBR (multi-source CBR) | $\sum_{i=1}^{k} w_i f_i(S_i)$ | The final prediction model is the weighted mean of different predictive models trained on the *k related* source areas. |
| Multiple source areas only with DA (multi-source DA) | $\frac{1}{N}\sum_{i=1}^{N} f_i(D_i)$ | The final prediction model is the average of predictions from all landslide models trained on the DA-*selected data* from different source areas. |
| Multiple source areas with CBR and DA (multi-source CBR-DA) | $\sum_{i=1}^{k} w_i f_i(D_i)$ | The final prediction model is the weighted mean of predictions from landslide models trained on the DA-*selected data* from the *k related* source areas. |
| **Benchmarks** | | |
| Multiple source areas without CBR and DA (multi-source transfer benchmark) | $\frac{1}{N}\sum_{i=1}^{N} f_i(S_i)$ | The final prediction model is the average of predictions from all landslide models trained on different source areas. |
| Single source area without CBR and DA (single-source transfer benchmark) | $f_i(S_i)$ | The final prediction model is trained on a (specific) single source area. |
| Target benchmark | $f(T)$ | The final prediction model is trained on data from the target area itself (only for comparison – not a model transfer situation). |

## 2.3 Susceptibility model training and testing

The transfer learning strategies were applied using generalized additive models (GAM) for susceptibility modelling. The logistic GAM, which performs a binomial classification of the absence/presence of landslide occurrence, has been well established as a method suitable for landslide susceptibility (Goetz et al., 2011; Conrad et al., 2015; Petschko, 2014; Bordoni et al., 2020). In fitting our model, we assume feature space is the same for source and target areas. We therefore only used common predictors of landslide susceptibility (Goetz et al 2015) that are available in all source and target areas, which include

local slope angle, plan and profile curvature, catchment slope angle, and upslope contributing area. These terrain attributes are intended to act as proxies for destabilizing forces (slope, catchment slope angle), water availability (logarithm of the size of the upslope contributing area, concave curvatures), and exposure to wind (convex curvatures), as well as general variability in characteristics of soil and vegetation (Muenchow et al., 2012). We used the mgcv package (Wood, 2006) for GAM implementation. We set the dimension of the basis used to represent the smooth term k as 4. Since it can be difficult to separate

landslide scarp and body from medium to low-resolution data (Dou et al., 2020), landslide presence points were randomly sampled from the entire landslide polygon and non-landslide points were randomly sampled from the area where the mapped landslides were excluded.

In turn, each study area was used as a target area. The landslide label data from the target area was not involved in the training process of all strategies embedded in CBR, DA or CBR-DA. Model performance was assessed using test data only within a

185 target area. The training data set was composed of an equal number of landslide and non-landslide samples. These landslide and non-landslide grid cells were obtained from the whole study area, or a subset of the study area based on DA.



Altogether, we explore five CBR and DA strategies for susceptibility modelling based on single and multiple sources areas, which are summarized in Table 2. In our implementation of CBR, only source areas related to the target area were used for modelling, where we defined related areas as source areas that had a (minimum) similarity score ≥ 0.65. In the case of DA

(without CBR), multi-source models were created for all $N$ source areas, excluding the target area. The final susceptibility models for multi-source CBR, DA and CBR-DA strategies were based on combing model predictions from multiple source areas (described in Table 2).

The area under the receiver operating characteristic (ROC) curve (AUROC) (Hosmer et al., 2013) was used to assess the predictive performance of the transferred models based on its predictions in the target area. In choosing the AUROC, we treat

model predictions as relative scores instead of actual probability estimates, which is common practice in landslide susceptibility modelling.

**2.4 Case study transfer source and target areas**

We demonstrate the application of CBR and DA for transfer learning using ten case study areas for source and target areas from three distinct geographic regions (Fig. 2): Ecuador (Reserva Biológica San Francisco (RBSF) area and highway) in the

Andes of Southern Ecuador (Brenning et al., 2015; Muenchow et al., 2012), Emilia Romagna Region (https://ambiente.regione.emilia-romagna.it) in northern Italy (Bologna, Modena, Parma, Piacenza, and Rimini) (Segoni et al., 2018; Piacentini et al., 2018; Rossi et al., 2010; Ciccarese et al., 2021), and Austria (Burgenland, Waidhofen and Paldau) (Petschko et al., 2012; Knevels et al., 2021; Knevels et al., 2019; Gasser et al., 2009). Rainfall is considered the main trigger of landslides in all study areas.

The study areas have the similar types of igneous rock (e.g., basalt), sedimentary rocks (e.g., sandstone) and metamorphic rock (e.g., schist), while RBSF has no igneous and sedimentary rocks. The above references provide additional detailed information of the study areas. We also summarised the geological information of all study areas in Table A1 in the *Appendix*.

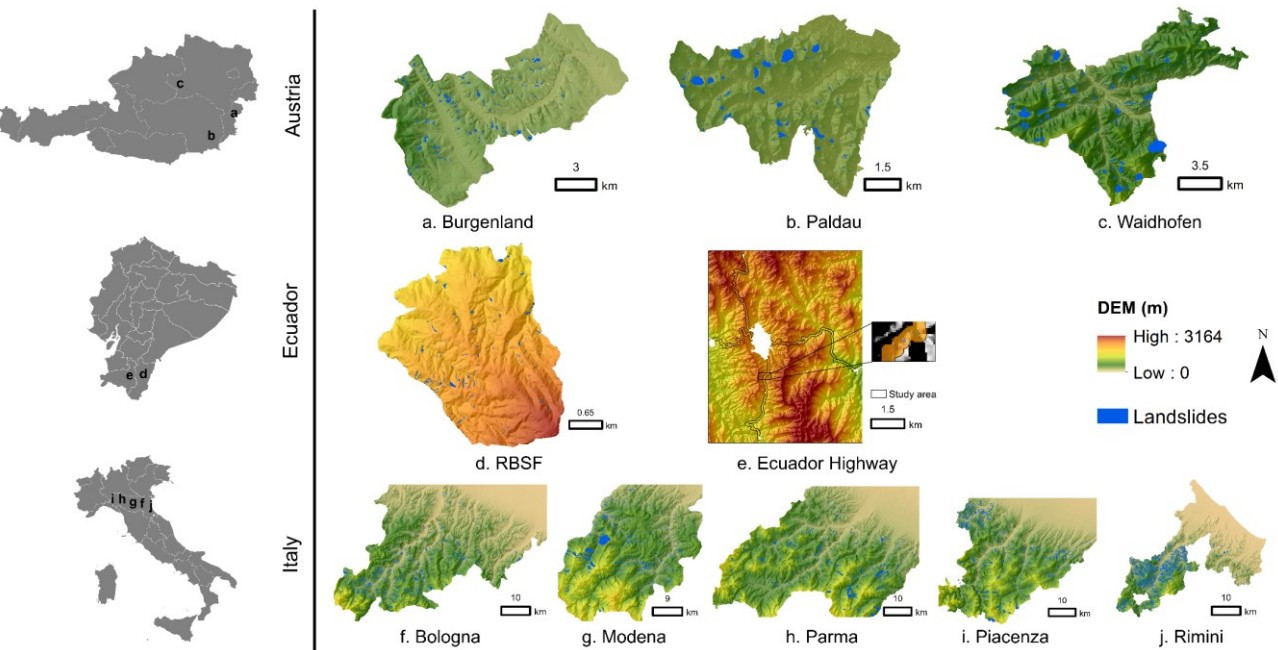

**Figure 2: Overview of study areas for landslide susceptibility mapping in our study. The study areas are shown as DEM map in the same scale. Landslide inventories of study areas are shown as blue polygons. From top to bottom, study areas are from Austria, Ecuador, and Italy. Map extents correspond to study areas, with the exception of the Ecuador highway area, where the study area is limited to a 300 m buffer on both sides of the highways and outside urban areas.**

In our study, DEMs with different resolutions were available for the Austrian, Italian, and Ecuadorian study areas. For Ecuador, 10 m × 10 m DEMs were produced by E. Jordan and L. Ungerechts (Düsseldorf), for Italy a EU-DEM with a 25 m × 25 m resolution was used (https://www.eea.europa.eu), and an airborne LiDAR-derived digital terrain model (DTM) with a 1 m × 1 m resolution was available for the Austrian areas from the GIS department of the Styrian government and the Government of Burgenland. Landslide inventories in our study were provided by J. Muenchow (Erlangen) for Ecuador, who also did a more detail study in Muenchow et al. (2012), SGSS (2019) for Emilia Romagna Region, Knevels et al. (2019) for Burgenland and Knevels et al. (2021) for Waidhofen and Paldau. For the Emilia Romagna Region, we chose the subset of landslides labelled as active.

We furthermore resampled the DEMs with 1 m resolution to 10 m and 25 m and the data with 10 m resolution to 25 m in order to use up to three dataset versions to mimic mismatches in target and source resolution. Resampling was based on B-Spline Interpolation in SAGA (System for Automated Geoscientific Analysis) GIS 7.4.0 (Conrad et al., 2015). Overall, we therefore had 17 datasets (Table 3). For brevity we combine the place name with the resolution, e.g., Waidhofen 10 for Waidhofen with a 10 m resolution.





**Table 3 Summary of the landslide data sets used in this study**

| Study Areas | Resolution(m) | Number of Landslide | Mean Landslide Size (m²) | Main landslide types | Main triggering factor | | Region |
|---|---|---|---|---|---|---|---|
| Burgenland | 1<br>10*<br>25* | 382 | 6330 | | | | Alpine fringe |
| Paldau | 1<br>10*<br>25* | 418 | 3879 | Earth and debris materials (Knevels et al., 2019; Knevels et al., 2020) | Rainfall | Austria | Styrian Basin |
| Waidhofen | 1<br>10*<br>25* | 621 | 11235 | | | | Ybbstaler Alps |
| RBSF | 10<br>25* | 178 | 733.9 | Shallow landslides (Muenchow et al., 2012) | Rainfall | Ecuador | South Ecuadorian Andes |
| Ecuador Highway | 10 | 1588 | 2725.4 | Shallow and deep-seated landslides (Brenning et al., 2015) | | | |
| Bologna<br>Modena<br>Parma<br>Piacenza<br>Rimini | 25 | 1354<br>1240<br>1261<br>1583<br>2229 | 33272<br>38816<br>41444<br>37502<br>34679 | Debris flows (Piacentini et al., 2018) | Rainfall and Earthquake | Italy | Italian Alps |

\* resolution of resampled data

## 3 Results

### 3.1 CBR similarity analysis

Mean slope angle and spatial resolution were for the majority the most limiting and therefore influential similarity attributes in determining which source areas were related to the target area based on the similarity score threshold of 0.65. For some target areas, multiple similarity attributes (i.e., topographic characteristics and geological unit) contributed to differentiate candidate source areas, while for others, a single attribute (mean slope or resolution) dominated the exclusion of unrelated source areas (Fig. 3). For high-resolution datasets, the resolution attribute was primarily responsible for the overall similarity.

The combination of spatial resolution and the standard deviation of slope or mean slope affected the overall similarity as the resolution between the source and target areas got closer. In general, as the spatial resolution of the target area and the source areas became coarser, the number of related source areas tended to increase. Mean slope, standard deviation of slope and geological units had more influence on the overall similarity decision making when resolutions were similar. Topographic characteristics and resolution were generally the main attributes that determined the overall similarity.

Most of the target areas (14 out of 17) had one or more related source area. There were only three cases where the target had no related source areas (RBSF 10, 25 and Waidhofen 1), two cases of only one source area, and twelve with multiple source areas (Fig. 4). Target areas with a resolution of 25m tended to have a high number of related source areas. Some target areas had related source areas in different geographic regions (e.g., Italian Alps and Burgenland).





Three representative target areas were selected to show contribution of each attribute to the overall similarity because similar

patterns were observed elsewhere (Figure 3; complete results in *Supplementary Material*).

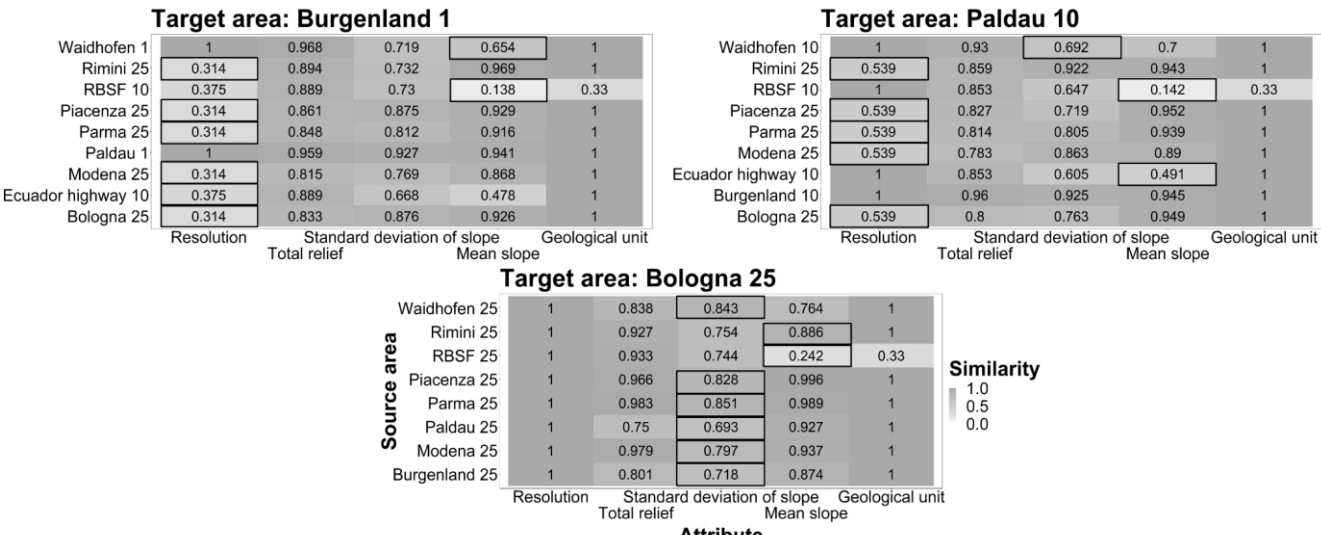

**Figure 3: Similarity scores for three selected representative target areas (Burgenland 1, Paldau 10 and Bologna 25). Light colours represent smaller similarities. The overall similarity value of each source area is marked with a black box.**

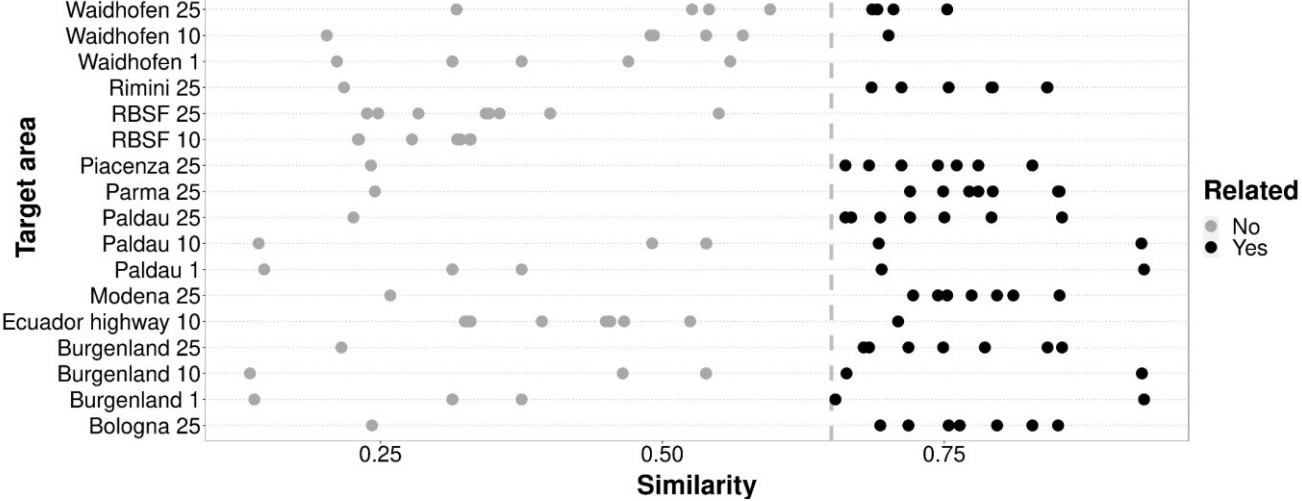

**Figure 4: Distribution of related (black points) and unrelated (grey points) source areas for different target areas in the CBR transfer learning strategies. Related source areas are defined as having a minimum similarity score ≥ 0.65 (vertical line).**

### 3.2 Single-source learning

In single-source transfer learning, CBR achieved the highest model performance overall. DA resulted in stronger predictive

performances only when source and target areas were substantially dissimilar (Fig. 5, 6). The AUROCs obtained by single-

255 source CBR were always distributed between the median and maximum values of transfer benchmark models and close to the





AUROCs obtained by the model trained using only target data (Fig. 5 and Table A2 in *Appendix*). E.g., when Bologna 25 was the test data, the AUROCs of a model trained by the most related source area Piacenza 25 was 0.762 and that of the model trained with Bologna 25 data itself was also 0.762. Moreover, the majority of median AUROC performances obtained with single-source DA were greater than the median AUROC performance of single-source transfer benchmark models (Fig. 5).

The distribution trend it displayed implied that single-source DA to some extent improved performances, which was consistent with the results shown in Fig. 6. Specifically, for similarities below 0.27, AUROCs achieved with DA were up to 0.14 higher than without it. When the overall similarity of the source area for the target area gradually increased up to ~0.60-0.65, the difference values were centred around +0.03. As the overall similarity was greater than 0.65, the AUROCs obtained by single-source DA were close to the ones achieved by the single-source transfer benchmark.

CBR-DA also showed good performance as its results were located in the upper part of transfer benchmark results (Fig. 5). This may be due to the contribution of CBR rather than DA. Throughout all the results, single-source CBR demonstrated more stable prediction performance behaviour compared to the results obtained by the strategies involving DA.

From this perspective, it can be concluded that by selecting the related areas, CBR was effective in identifying a suitable source area that results in favourable performances regardless of the use of DA.

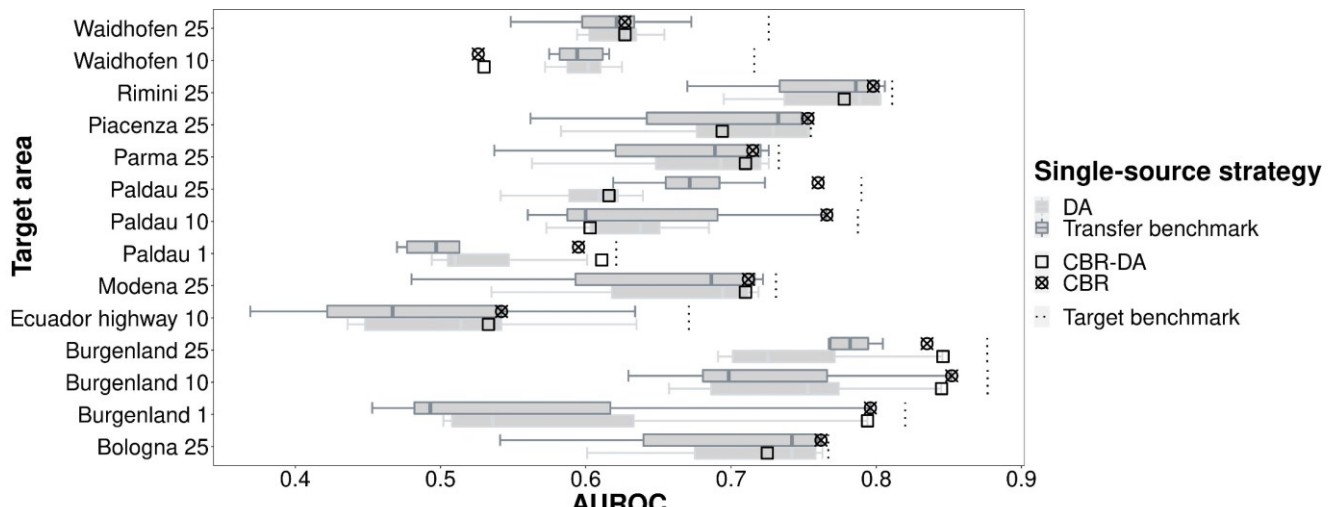

**Figure 5: Comparison of single-source strategies: AUROCs obtained by models trained on individual source areas with case-based reasoning (CBR), domain adaptation (DA), combined CBR-DA, and in the single-source transfer benchmark.**



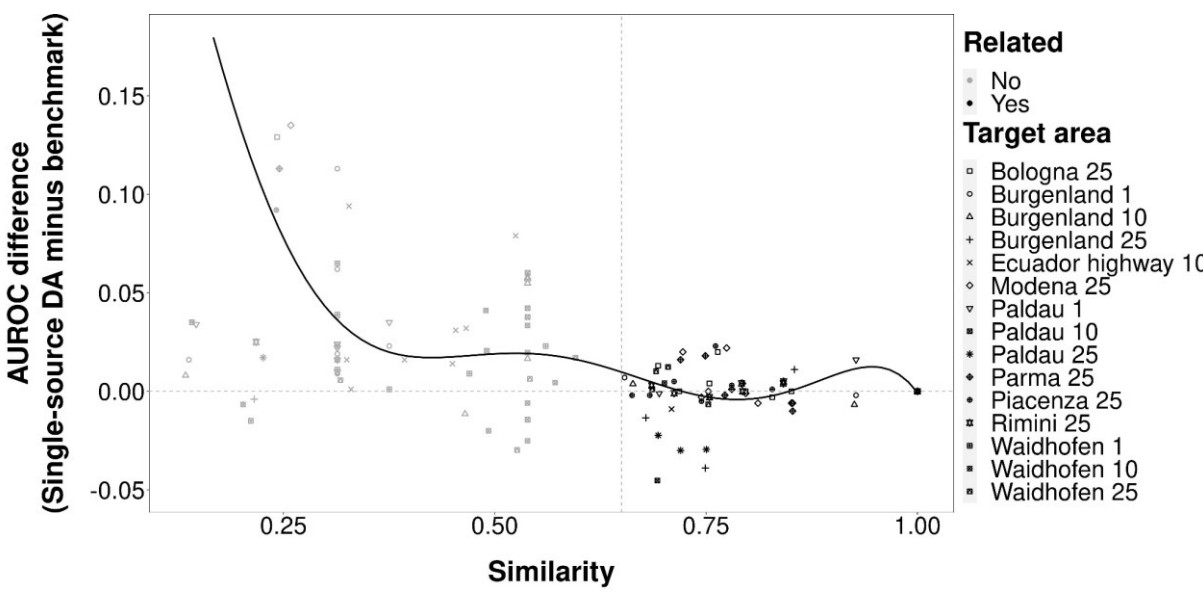

**Figure 6: Similarity scores vs. AUROC differences between models trained on individual source areas with domain adaptation and models without domain adaptation ("single-source DA" minus "single-source transfer benchmark").**

### 3.3 Multi-source learning

The strategies that involved CBR had better prediction performances in the multi-source transfer learning in comparison to the multi-source transfer benchmark and multi-source DA (Fig. 7).

Multi-source CBR obtained good performances regardless of the number of related source areas and whether the related source areas were from the same region (Fig. 4 and Fig. 7). However, multi-source CBR-DA in general underperformed, usually having predictive performance lower than the multi-source transfer benchmark. When comparing the average of AUROCs of different strategies for all target areas in the multi-source transfer learning, CBR was the best multi-source strategy followed by the transfer benchmark, and CBR-DA, while DA had the worst multi-source performance. Furthermore, with respect to the stability of the results, multi-source CBR performed best since the performances it obtained were always in the top two of all performances obtained by different multi-source transfer learning strategies. In contrast, the results obtained for strategies involving DA were highly variable and always inferior to the results of the corresponding multi-source transfer benchmark.





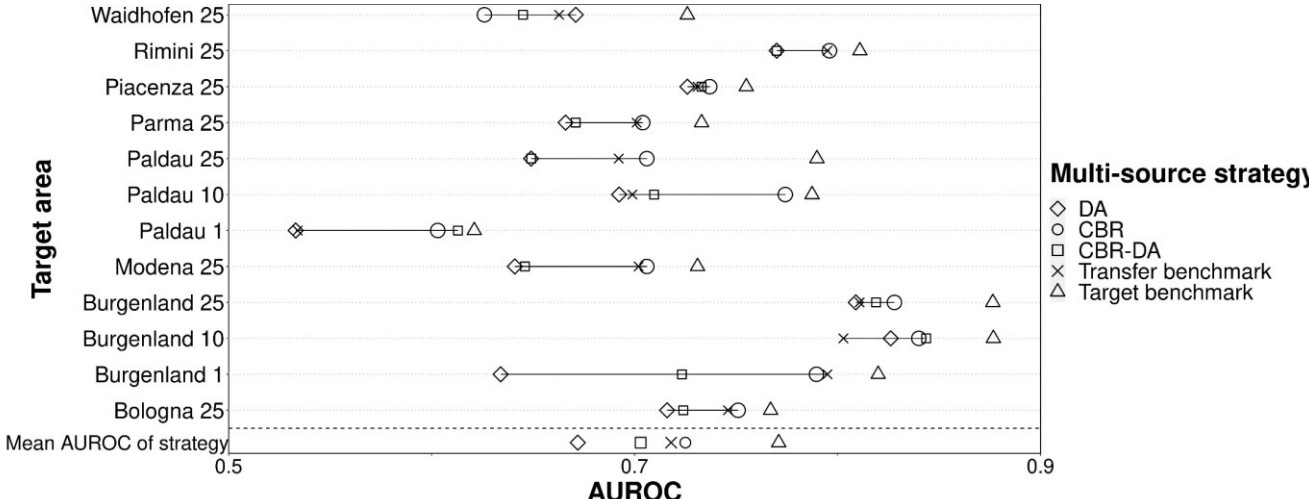

**Figure 7: AUROCs of models trained on multiple source areas with case-based reasoning (CBR), domain adaptation (DA), combined CBR-DA, and the multi-source transfer benchmark (averaged across all source areas) and target benchmark.**

## 3.4 Comparing single- and multi-source learning

Single-source CBR was for the majority the best-performing transfer learning strategy, closely followed by multi-source CBR (Fig. 8). Both were located between the median and the maximum of single-source transfer benchmark and tended to be closer to the maximum, which means that CBR-based source selection was highly effective at identifying the most suitable sources of training data. On average the single-source and multi-source CBR AUROCs were below the overoptimistic target benchmark (training and testing in target area) by only ~0.05. The strong performance of CBR in both single- and multi-source strategies indicates that the most effective transfer learning methods were to train the predictive model using the most related source area or performing a weighted combination based on the similarity scores of the predictive models trained on the most strongly related source areas.





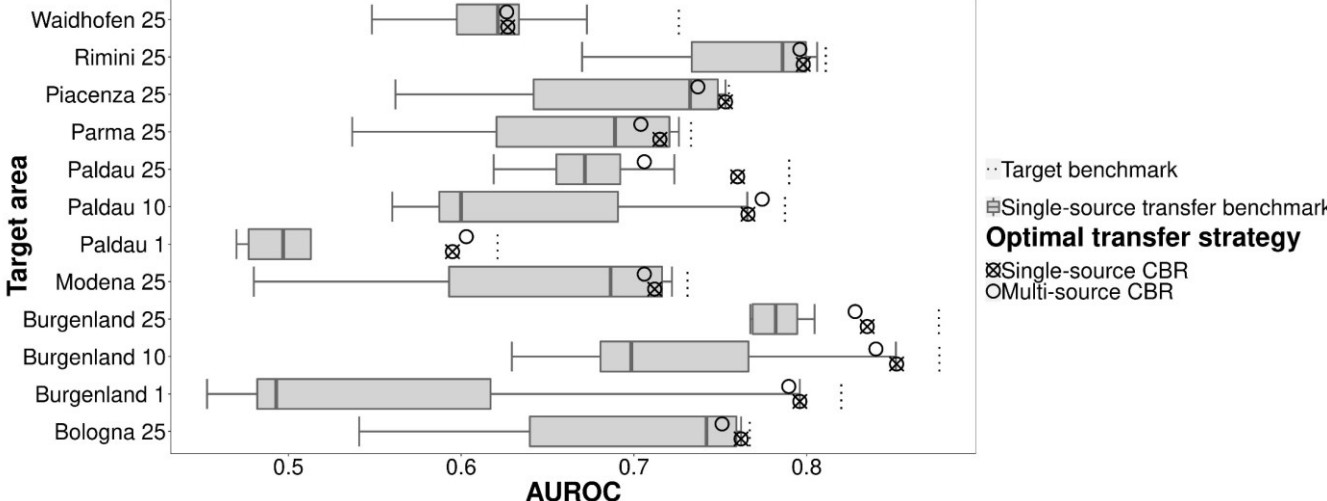

**Figure 8: Comparison of single- and multi-source CBR strategies and the single-source target benchmark.**

## 3.5 Comparing susceptibility map appearances

The highest performing transfer learning strategies (single-source and multi-source CBR and CBR-DA) had spatial patterns of landslide prone areas that most resembled the target benchmark (Fig. 9). Strategies with CBR, which considers target-source similarity, were able to better avoid falsely detecting landslide prone areas. Using classified landslide susceptibility maps for

Burgenland 10 as an example, the lower performing multi-source DA (Fig. 9h) and the multi-source benchmark (Fig. 9e) appear to overpredict susceptibility in some areas (e.g., on alluvial fans) compared to the target benchmark (Fig. 9a) and the higher performing CBR-based transfer learning strategies (Fig. 9c, f, and g). The susceptibly maps also show that given a single source area has a high similarity (e.g., Paldau 10 and Burgenland 10) to the target area, DA strategies (Fig. 9d) can also properly detect landslide susceptible areas. The difference in landside prone areas of single- and multi-source benchmarks

compared to the target benchmark also indicates that not all source areas were suitable for predicting landslides for unseen areas.

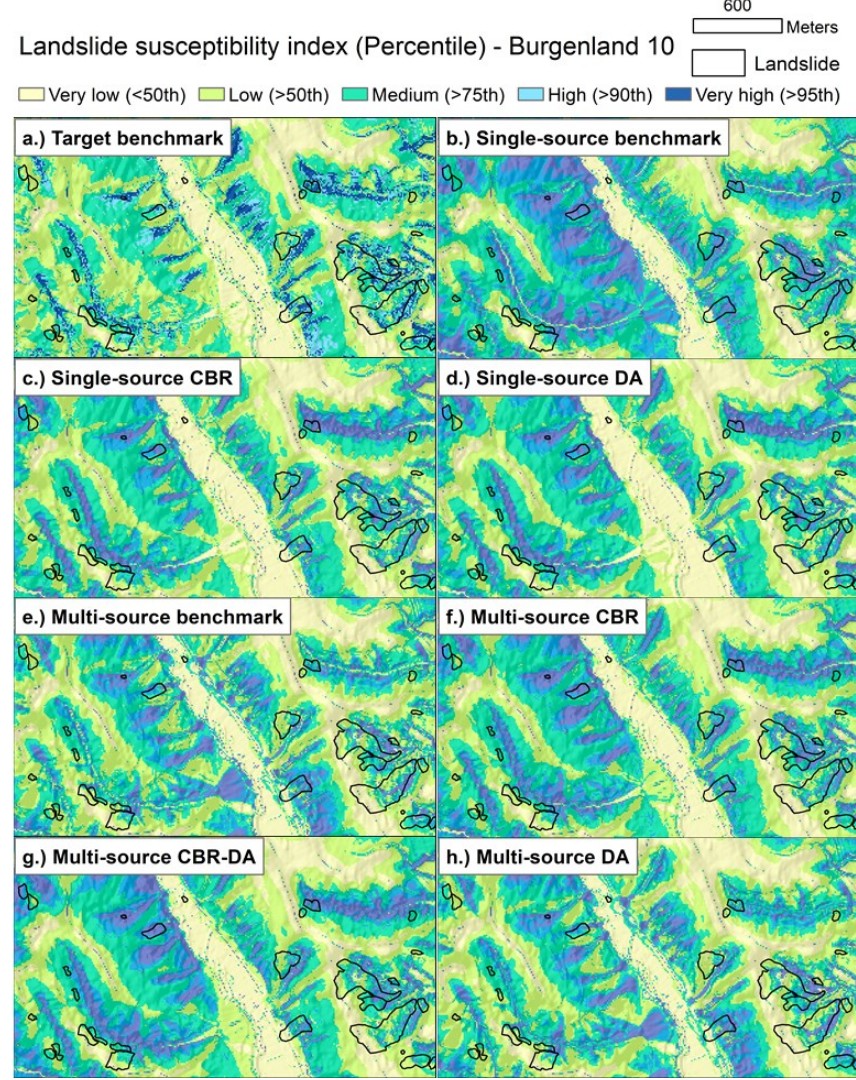

**Figure 9: An example of classified landslide susceptibility maps for each benchmark and transfer learning strategy for Burgenland 10. Predicted probabilities are classified into five susceptible levels (very high, high, moderate, low, very low) using the top 5th, 10th, 25th and 50th percentile of each individual strategy's prediction. The results of single-source CBR and single-source DA are illustrated using models trained by Paldau 10. The single-source benchmark result is illustrated using model trained by Waidhofen 10.**

## 4 Discussion

### 4.1 Case-based reasoning in landslide assessment studies

By calculating the similarities between source and target areas to find the most transferable or one of the most transferable source areas, CBR is able to transfer the knowledge from source areas to the target area. In our study, we considered the data from a variety of different regions, and our results provided a more comprehensive understanding of the potential of CBR in





single- and multi- source transfer learning. Consistent with the literature for digital soil mapping and digital terrain analysis (Liang et al., 2021; Liang et al., 2020a; Qin et al., 2016), our results further support the adoption of CBR and provide useful methodological information for landslide assessment studies.

CBR may give a fresh insight to improve the understanding of knowledge transfer in landslide susceptibility modelling. It is an effective method to capture past experiences, intensifying and enriching the knowledge to improve the predictive

capabilities of models (Bannour et al., 2021; Wang et al., 2020). Particularly, it only needs to consider the basic characteristics of the data and the region to quickly match historical scenarios to the current study area and thus solve the task at hand. Meanwhile, the use of CBR to compare similarities between datasets makes it possible to reuse existing predictive models. These attractive abilities may benefit ex-post landslide mapping for emergency response as well as ex-ante landslide susceptibility modelling for hazard mitigation. Moreover, we determined that using similarity as the basis for the weight of

each related source area and the strategies involved CBR in multiple source areas displayed good and robust performance in our study (Fig. 7).

Until now, model transfer in landslide modelling have usually relied on a homogeneous availability of data and a strong model generalization to avoid local overfitting and allow the application of a model in an adjacent target region (Goetz et al., 2011; Wenger and Olden, 2012; Petschko et al., 2014; Bordoni et al., 2020). Although this approach has been identified as a robust

method for regional susceptibility modelling, its model transferability is often limited to nearby locations that have the same feature space and a nearly identical data distribution. However, when data distribution is different, the above approach may not be effective, even though the training data are from adjacent regions. Yates et al. (2018) have pointed out that the spatial and temporal separation may have little impact on model transfers, while environmental dissimilarity and data resolution are critical factors for successful model transfer. These factors could be considered as the spatial and temporal limits to

extrapolation in model transfers, as well as for landslide susceptibility model transfers. CBR may be able to handle these limits by calculating the overall similarity, indicating the likelihood of implementing landslide susceptibility model transfers between different study areas. In Fig. 3 and 7 we found that combining data from multiple related source areas with CBR obtained great results, even though some of the related source areas are from different regions as the target area.

After selecting related source areas, the predictors designed for training the model need to be examined. In our study, we

assumed that the source and target areas used the same predictors and focused on topographic predictors. However, when the source and target areas have different predictors, one of the problems is that topographic predictors are not the only factors that play a key role in landslide prediction. Thus, a method should be implemented to select suitable predictors for model transferring since not all predictors can be used in the training process – Liang et al. (2021) selected suitable predictors for a new task by using each model trained by individual predictors of the source area to predict in the target area and concluded

this method was effective. However, since they only focused on terrain attributes, it is unclear how this approach would work on other predictors such as antecedent rainfall intensity, which, in addition to regional rainfall pattern variations, can strongly



differ from one region to another. From this perspective, we would suggest future research using CBR transfer learning could focus on the selection of features that are more likely transferable.

## 4.2 CBR Similarity criteria

The proposed similarity scores in this study based on geologic, topographic and data characteristics (i.e., spatial resolution) worked quite well at supporting CBR strategies for identifying the most similar and thus transferable source areas. These similarity attributes do not explicitly account for landslide type, which is an important factor to consider when landslide susceptibility modelling (Huang and Zhao, 2018). However, geologic attributes and terrain attributes such as slope angle, may work together as a suitable surrogate to anticipate the most likely landslide types given little to no landslide data in the target

area. Landslide type information is also difficult to collect and is often lacking in landslide inventories (Mezaal and Pradhan, 2018). Prior information on unseen areas or integrating expert experience may be helpful in formulating landslide types for transfer learning.

In general, the use of similarity indices can be somewhat arbitrary, as there are currently no clear criteria for how to select suitable similarity indices. For example, Liang et al. (2020a) analysed the importance of each attribute for digital soil mapping

based on previous studies to select the similarity index. Qin et al. (2016) indicated that the similarity indices should be structured to effectively represent the contextual information relevant to digital terrain analysis applications, hence the similarity indices used were based on knowledge and experience. Wang et al. (2020) selected similarity indices based on their importance for disaster situations.

For CBR applied to landslide susceptibility modelling, more elaborate criteria that could be indirectly used to account for

differences in landslide type could focus on triggering conditions such as land use (Knevels et al. (2021) and Steger et al. (2017)), distance to paved and unpaved road networks; noting that (paved) roads in Ecuador are associated with strongly increased landslide susceptibility (Brenning et al., 2015). Adding more process-related similarity indices may lead to improved CBR transfer learning, but this may not be easy to implement across different study regions in different countries with different mapping agencies and standards. Therefore, similar to selecting individual features for landslide susceptibility modelling

(without transferring), we recommend the use of expert knowledge to help guide in the selection of similarity attributes.

In terms of choosing related source areas, the minimum operator method worked very well in our study and avoided selecting a "false" related source. However, we did observe a scenario where one area was considered related but the reciprocal area not (Paldau and Waidhofen; Fig. 4). As pointed out by Humphreys et al. (2003), when using CBR for similarity evaluation, the evaluation criterion used may be different in different categories and situations. By analogy, we can assume that the threshold

settings for similarity may also differ for different attributes in different study areas in landslide assessment studies. Additionally, there are other methods to obtain the related area, such as Manhattan distance, grey relational analysis, or $k$-nearest neighbours (Dou et al., 2015).



### 4.3 Utility of domain adaptation in geospatial learning

Our study showed that DA did not generally improve transfer learning performance in landslide susceptibility modelling. This
holds true for single-source as well as multi-source DA with and without CBR-based source selection. Nevertheless, DA
increased the AUROC performance when the source area was rather dissimilar to the target area (Fig. 6), which is less relevant
in landslide studies that have access to a large and geographically diverse case base. It is impressive that models trained on
multiple related source areas with CBR and DA showed good performances. For instance, when Paldau with a 1 m × 1 m, and
Burgenland with a 10 m × 10 m resolution were the target areas, AUROCs obtained by multi-source CBR-DA were nearly
395 equal to that obtained by the best single-source transfer benchmark and higher than the other strategies (Fig. 7). The reason
may lie in the improvement of DA through the weighting of source areas.

A further consideration is to use labelled data from the target area. Fang et al. (2021) proposed a new domain adaptation for
landslide inventory mapping by considering pre-landslides and post-landslide conditions and concluded that the proposed
method was successful. This new method could be considered as supervised DA in landslide susceptibility mapping. In other
geospatial learning fields, such as land cover mapping, Mboga et al. (2021) compared two unsupervised domain adaptation
strategies (the correlation alignment (D-CORAL) domain adaptation network and the domain adversarial neural, DANN) and
found that classification performance was improved by adding labelled data from the target area. We suggest that active-
learning strategies (Wang & Brenning, 2021) could be useful in generating limited amounts of labelled data for transfer
learning.

**5 Conclusion**

The aim of our study was to examine the performances of geographically informed case-based reasoning (CBR) and
unsupervised domain adaptation (DA) in geographically transferring knowledge for landslide susceptibility modelling in
"new" target areas without landslide inventory data. Our comparative study revealed that CBR strategies with a single source
area as well as with multiple related source areas are robust and effective in developing highly transferable landslide
susceptibility models without any a prior knowledge of landslides in target area. In the multi-source CBR and CBR-DA transfer
strategies the weighted combination of model predictions based on similarity scores can qualitatively exploit the degree of
relatedness between target and source areas, but the way to determine the similarity still requires further research. Single-
source CBR was the most effective method for performing model transfer to the target area in most situations. Its performance
was also very close to that obtained by models trained with data from the target area itself. Domain adaptation showed promise
for situations where only source areas that are weakly related to the target area are available for transfer learning. Nevertheless,
considering the increasing availability of geographically diverse landslide inventories, this appears to be of limited practical
relevance in landslide studies. Therefore, the single-source CBR strategy appears to be the most promising strategy for
landslide susceptibility modelling.





The findings of this paper also provide insights regarding the potential of transferring existing landslide susceptibility models
to new areas. By calculating the similarity between data and region characteristics, trained models can directly be used for the
new task, especially in situations that require rapid model development, such as emergency situations. Although further
research is needed to generalize our findings, we suggest that the proposed approach can alleviate the burden of collecting and
labelling data, resulting in a more expedited preparation of landslide susceptibility maps for large and data-scarce regions.

**Appendix**

**Table A1 Information of all study areas**



| | | Original dataset | | | | | | | | | |
|---|---|---|---|---|---|---|---|---|---|---|---|
| | | **Burgenland** | **Paldau** | **Waidhofen** | **Ecuador highway** | **RBSF** | **Modena** | **Parma** | **Piacenza** | **Rimini** | **Bologna** |
| Slope angle (°) | Min | 0 | 0 | 0 | 2.5 | 0 | 0 | 0 | 0 | 0 | 0 |
| | Max | 82.5 | 66.6 | 87.5 | 52.9 | 76.2 | 67.8 | 71.8 | 66.6 | 64.1 | 67.2 |
| | Mean | 9.2 | 10.9 | 19.8 | 25.3 | 35.7 | 13.3 | 11.8 | 11.4 | 8.2 | 11.5 |
| | Standard deviation | 8.2 | 8.3 | 10.7 | 12.2 | 10.7 | 6.9 | 7.5 | 8.6 | 6.5 | 7.9 |
| Area (km²) | | 117.8 | 39.3 | 131.3 | 88 | 9.6 | 1293 | 2576.8 | 1834.8 | 921.2 | 3707.6 |
| Main geological units | Igneous | ✓ | ✓ | ✓ | ✓ | ✗ | | | ✓ | | |
| | Sedimentary | ✓ | ✓ | ✓ | ✓ | ✗ | | | ✓ | | |
| | Metamorphic | ✓ | ✓ | ✓ | ✓ | ✓ | | | ✓ | | |
| Elevation (m) | Max | 766.7 | 463.1 | 1114.9 | 2960.4 | 3164.2 | 2141.5 | 1827.3 | 1726.3 | 1399.3 | 1923.5 |
| | Min | 243.1 | 282.7 | 324.5 | 948.8 | 1714.4 | 80.7 | 34.8 | 49.7 | 3.8 | 11.0 |
| | | Predictor variables for landslide and non-landslide observations | | | | | | | | | |
| Slope angle (°) | Landslides median (IQR) | 15.9 (13.95) | 11.36 (11.63) | 19.05 (13.84) | 30.02 (17.43) | 43.37 (10.67) | 12.08 (5.64) | 11.03 (6.49) | 10.81 (5.83) | 10.17 (4.93) | 11.2 (5.5) |
| | Non-landslides median (IQR) | 6.7 (8.24) | 9.23 (10.53) | 18.00 (15.12) | 23.03 (17.56) | 36.27 (14.59) | 12.34 (8.07) | 11.30 (9.74) | 10.31 (1.08) | 7.18 (9.88) | 10.9 (10.1) |
| Plan curvature (radians per 100 m) | Landslides median (IQR) | -0.001 (0.33) | 0.00136 (0.39) | -0.0034 (0.3648) | 0.00086 (0.0152) | -0.017 (0.052) | 0.00124 (0.00804) | 0.00102 (0.00785) | -0.00182 (0.0081) | 0.00214 (0.00813) | -0.002 (0.0079) |
| | Non-landslides median (IQR) | 0.00019 (0.42) | 0.00294 (0.46) | -0.0009 (0.2762) | 0.00028 (0.0117) | 0.0054 (0.043) | 0.00035 (0.00873) | 0.00037 (0.00935) | 0.00027 (0.00999) | 0.00052 (0.01118) | 0.0004 (0.01) |
| Profile curvature (radians per 100 m) | Landslides median (IQR) | -0.0001 (0.0014) | 0.00025 (0.08558) | -0.0015 (0.1028) | 0.00056 (0.019) | -0.0025 (0.013) | 0.00015 (0.0015) | 0.00014 (0.00143) | -0.00019 (0.00135) | 0.00018 (0.00127) | -0.0002 (0.0014) |
| | Non-landslides median (IQR) | 0.00024 (0.05) | 0.00046 (0.07183) | -0.0006 (0.0738) | -0.00036 (0.013) | 0.00147 (0.014) | 0.00002 (0.0017) | 0.00002 (0.0014) | -0.00002 (0.00132) | 0.00002 (0.00101) | -3e-05 (0.0015) |
| Upslope contributing area (log₁₀ m²) | Landslides median (IQR) | 1.74 (1.08) | 1.56 (0.95) | 1.87 (0.997) | 3.04 (0.65) | 3.03 (0.61) | 4.19 (0.76) | 4.16 (0.71) | 4.18 (0.73) | 4.07 (0.72) | 4.1 (0.71) |
| | Non-landslides median (IQR) | 1.67 (0.97) | 1.56 (0.95) | 1.88 (0.734) | 3.15 (0.80) | 2.80 (0.60) | 3.86 (0.67) | 3.81 (0.70) | 3.82 (0.74) | 3.71 (0.68) | 3.7 (0.67) |





**Table A2 AUROCs of models trained on individual source areas with domain adaptation versus without domain adaptation, the results are shown as DA / target benchmark. The bolded font indicated that the source area corresponding to this AUROC was the most related for the current target area.**

| | Target areas | | | | | | | | | |
|---|---|---|---|---|---|---|---|---|---|---|
| | Bologna 25 | Burgenland 1 | RBSF 10 | Ecuador highway 10 | Modena 25 | Paldau 1 | Parma 25 | Piacenza 25 | Rimini 25 | Waidhofen 1 |
| Bologna 25 | 0.767 | 0.515/0.432 | 0.524/0.505 | 0.446/0.415 | 0.719/0.72 | 0.494/0.47 | 0.72/0.726 | **0.754/0.753** | 0.803/0.806 | 0.5/0.435 |
| Burgenland 1 | - | 0.82 | - | - | - | **0.611/0.595** | - | - | - | 0.562/0.553 |
| RBSF 10 | - | 0.633/0.617 | 0.772 | 0.635/0.634 | - | 0.547/0.513 | - | - | - | 0.543/0.558 |
| Ecuador highway 10 | - | 0.536/0.628 | 0.704/0.713 | 0.671 | - | 0.505/0.47 | - | - | - | 0.502/0.501 |
| Modena 25 | 0.756/0.756 | 0.502/0.493 | 0.468/0.406 | 0.448/0.369 | 0.731 | 0.512/0.502 | **0.705/0.715** | 0.743/0.748 | **0.802/0.798** | 0.496/0.48 |
| Paldau 1 | - | **0.794/0.796** | - | - | - | 0.621 | - | - | - | 0.587/0.564 |
| Parma 25 | **0.762/0.762** | 0.504/0.466 | 0.629/0.594 | 0.514/0.482 | **0.706/0.712** | 0.499/0.477 | 0.733 | 0.753/0.75 | 0.795/0.795 | 0.496/0.473 |
| Piacenza 25 | 0.757/0.76 | 0.595/0.482 | 0.54/0.428 | 0.436/0.422 | 0.719/0.722 | 0.51/0.494 | 0.721/0.72 | 0.755 | 0.803/0.804 | 0.499/0.488 |
| Rimini 25 | 0.763/0.759 | 0.508/0.489 | 0.681/0.552 | 0.483/0.467 | 0.709/0.715 | 0.507/0.497 | 0.726/0.722 | 0.753/0.748 | 0.811 | 0.533/0.494 |
| Waidhofen 1 | - | 0.77/0.763 | - | - | - | 0.601/0.602 | - | - | - | 0.652 |
| Burgenland 10 | - | - | 0.657/0.533 | 0.542/0.448 | - | - | - | - | - | - |
| Paldau 10 | - | - | 0.73/0.73 | 0.63/0.614 | - | - | - | - | - | - |
| Waidhofen 10 | - | - | 0.709/0.662 | **0.533/0.542** | - | - | - | - | - | - |
| Burgenland 25 | 0.728/0.728 | - | - | - | 0.683/0.661 | - | 0.681/0.663 | 0.715/0.717 | 0.782/0.777 | - |
| Paldau 25 | 0.601/0.588 | - | - | - | 0.535/0.515 | - | 0.563/0.547 | 0.583/0.585 | 0.715/0.711 | - |
| Waidhofen 25 | 0.677/0.657 | - | - | - | 0.619/0.619 | - | 0.643/0.645 | 0.684/0.661 | 0.744/0.741 | - |
| RBSF 25 | 0.67/0.541 | - | - | - | 0.615/0.48 | - | 0.65/0.537 | 0.654/0.562 | 0.695/0.67 | - |

| | Burgenland 10 | Paldau 10 | Waidhofen 10 | Burgenland 25 | Paldau 25 | Waidhofen 25 |
|---|---|---|---|---|---|---|
| Bologna 25 | 0.686/0.629 | 0.5/0.560 | 0.608/0.575 | 0.705/0.768 | 0.639/0.662 | 0.627/0.617 |
| RBSF 10 | 0.774/0.766 | 0.726/0.691 | 0.587/0.594 | - | - | - |
| Ecuador highway 10 | 0.657/0.669 | 0.638/0.617 | **0.530/0.526** | - | - | - |
| Modena 25 | 0.753/0.736 | 0.642/0.6 | 0.602/0.582 | 0.722/0.784 | 0.615/0.645 | **0.594/0.601** |
| Parma 25 | 0.510/0.696 | 0.651/0.591 | 0.610/0.616 | 0.728/0.767 | 0.589/0.619 | 0.654/0.642 |
| Piacenza 25 | 0.735/0.681 | 0.61/0.572 | 0.572/0.597 | 0.569/0.791 | 0.541/0.659 | 0.632/0.631 |
| Rimini 25 | 0.756/0.698 | 0.573/0.587 | 0.625/0.584 | 0.691/0.780 | 0.588/0.682 | 0.632/0.626 |
| Burgenland 10 | 0.877 | **0.603/0.766** | 0.596/0.616 | - | - | - |
| Paldau 10 | **0.845/0.852** | 0.787 | 0.616/0.612 | - | - | - |
| Waidhofen 10 | 0.807/0.803 | 0.6845/0.73 | 0.716 | - | - | - |
| Burgenland 25 | - | - | - | 0.876 | **0.616/0.760** | 0.643/0.673 |
| Paldau 25 | - | - | - | **0.846/0.835** | 0.790 | 0.605/0.588 |
| Waidhofen 25 | - | - | - | 0.791/0.805 | 0.603/0.723 | 0.726 |
| RBSF 25 | - | - | - | 0.765/0.769 | 0.699/0.681 | 0.554/0.548 |





**Code availability**

The scripts of strategies used in our paper are available at https://doi.org/10.5281/zenodo.6527716 (last access: 07 May 2022).

**Data availability**

**Austrian study areas:** Landslide inventories for Paldau and Waidhofen is available in Knevels et al., 2021 (https://doi.org/10.3390/land10090954) and for Burgenland is available in Knevels et al., 2019 (https://doi.org/10.3390/ijgi8120551). LiDAR-based HRDTM of Burgenland, Paldau and Waidhofen can be requested from the GIS department of the Styrian and the Government of Burgenland, and the provincial government of Lower Austria, respectively.

**Italian study areas:** Emilia Romagna Region landslide inventories could be downloaded at https://ambiente.regione.emilia-romagna.it/it/geologia/cartografia/webgis-banchedati/cartografia-dissesto-idrogeologico#consulta-dati-shp. The DEM for Emilia Romagna Region is available at https://www.eea.europa.eu/data-and-maps/data/copernicus-land-monitoring-service-eu-dem.

**Ecuadorian study areas:** Landslide data for the RBSF area is available as part of the open-source 'sperrorest' package in R (https://cran.r-project.org/package=sperrorest, dataset 'ecuador'), and the Ecuador highway landslide data is available from A. Brenning upon request. The DEMs used can be requested from the DFG Research Unit FOR 816 (J. Bendix, University of Marburg, Germany).

**Author contributions**

The conceptualization and methodology of the research was developed by ZW, JG and AB. The coding scripts that configured the data for training and testing were written by ZW. The analysis and interpretation of the data were carried out by ZW, JG and AB. The original draft of the paper was written by ZW, with edits, suggestions, and revisions provided by AB and JG.

**Competing interests**

The authors declare that they have no conflict of interest.

**Acknowledgements**

We thank the DFG Research Unit FOR 816 (J. Bendix, Marburg) for providing the Ecuadorian DEMs created by E. Jordan and L. Ungerechts, Düsseldorf. We are also grateful to the Federal State of Burgenland, the GIS department of the Styrian




government and the Provincial Government of Lower Austria for providing the high-resolution DEM for the Austrian study areas. Zhihao Wang was funded through a China Scholarship Council PhD scholarship, which is gratefully acknowledged.

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
