# Peer review of "Transfer learning for landslide susceptibility modelling using domain adaptation and case-based reasoning"

_Geoscientific Model Development, 2022_

## Referee Comment (RC1)

Dear Editor:

Thank you for giving me the opportunity to revise the manuscript entitled "Transfer learning for landslide susceptibility modelling using domain adaptation and case-based reasoning" by Zhihao Wang and his/her colleagues that was submitted to "Geoscientific Model Development". The manuscript used transfer-learning strategies to landslide susceptibility assessment (LAS) for 10 different cases in Austria, Ecuador, and Italy. Application of various learning based algorithms are interesting topics in LSA, but regarding the existed literatures needs to be polished properly. In this regard, the following comments are requested to be addressed by the authors:

Comment 1: The English of the paper is readable; however, I would suggest the authors to have it checked preferably by a native English-speaking person to avoid any mistakes.

Comment 2: Add the "....several case studies" in the title.

Comment 3: The concluding remarks of the abstract are not well-written. It's merely the repetition of the objectives and title of the manuscript. in a appropriate abstract, quantitative finding, method limitations, model verification, learning rate and justification have to added. Please kindly modify the abstract.

Comment 4: Considering the existing literature, the motivation of the manuscript is not convincing. There is a large literature associated with LSA, and various techniques have been applied/developed to assess the susceptibility better. I expect to see a novelty in a manuscript dealing with landslide susceptibility because we already have many case studies. I do not think that new case studies would help the scientific community to go one step

further. The result of this manuscript is only valid for the examined area. Based on site-specific conditions and the quality of a landslide inventory, different results can be obtained in another site, and another technique may appear like the best alternative. However, in the end, these efforts do not help us decide on the best method of LSA. So, please kindly request to provide solid document to support the superiority of the method than other approaches. Please check and refer these papers:

1. Kavzoglu, T., Colkesen, I., & Sahin, E. K. (2019). Machine learning techniques in landslide susceptibility mapping: a survey and a case study. Landslides: Theory, practice and modelling, 283-301.

2. Saleem, N., Huq, M., Twumasi, N. Y. D., Javed, A., & Sajjad, A. (2019). Parameters derived from and/or used with digital elevation models (DEMs) for landslide susceptibility mapping and landslide risk assessment: a review. ISPRS International Journal of Geo-Information, 8(12), 545.

Comment 5: The necessity & novelty of the manuscript should be presented and stressed in the "Introduction" section. Please check and refer these papers:

1. Nanehkaran, Y. A., Mao, Y., Azarafza, M., Kockar, M. K., & Zhu, H. H. (2021). Fuzzy-based multiple decision method for landslide susceptibility and hazard assessment: A case study of Tabriz, Iran. Geomechanics and Engineering, 24(5), 407-418.

2. Ngo, P. T. T., Panahi, M., Khosravi, K., Ghorbanzadeh, O., Kariminejad, N., Cerda, A., & Lee, S. (2021). Evaluation of deep learning algorithms for national

scale landslide susceptibility mapping of Iran. Geoscience Frontiers, 12(2), 505-519.

Comment 6: Provide a literature of the methods used learning-based LSA methods in "Introduction". The use of a table to demonstrate the advantage-disadvantage of these methods can be useful. Towards the end, mention the superiority & repeat the novelty of your work. Please check and refer these papers:

1. Wang, Y., Fang, Z., & Hong, H. (2019). Comparison of convolutional neural networks for landslide susceptibility mapping in Yanshan County, China. Science of the total environment, 666, 975-993.

2. Fang, Z., Wang, Y., Peng, L., & Hong, H. (2020). Integration of convolutional neural network and conventional machine learning classifiers for landslide susceptibility mapping. Computers & Geosciences, 139, 104470.

Comment 7: A relevant source of subjectivity and uncertainty is introduced when splitting the input parameters into an arbitrary number of classes with random break values. These choices affect the results. Would you please describe your solution?.

Comment 8: The methodology section is weakly written. So, my suggestion is to reconstruct it. In addition, please kindly use the flowcharts, model verification, benchmarks and error tables to provide detailed methodology.

Comment 9: What will be happened if you use this algorithm for another region?.

Comment 10: Please add a subsection clearly articulating the main limitations, wider applicability of your methods, and findings in the "Discussion" section. Also, the authors should deepen the discussion. Please check and refer these papers:

1. Wu, Y., Ke, Y., Chen, Z., Liang, S., Zhao, H., & Hong, H. (2020). Application of alternating decision tree with AdaBoost and bagging ensembles for landslide susceptibility mapping. Catena, 187, 104396.

2. Azarafza, M., Ghazifard, A., Akgün, H., & Asghari-Kaljahi, E. (2018). Landslide susceptibility assessment of South Pars Special Zone, southwest Iran. Environmental Earth Sciences, 77(24), 1-29.

Comment 11: Please define susceptibility as this terminology is poorly understood by many researchers.

Comment 12: There are no LSA maps that indicate the degree of the susceptibility, please add the required map from triggering factors to risk map. In LSA analysis, providing maps is very important to understand the authors' results.

Comment 13: Please provide justifications for your model, I believe application of other learning based method can be providing much better results (for example: deep learning). Please verify your model with well-known methods.

Comment 14: What type landslide was analyzed? Needs to describe.

Comment 15: Please add the "case study" section and provide relevant information about the geology, morphology, seismicity, hydrology and other triggering factors that impact on LSA.

Comment 16: Please provide the table of hyper-parameters values of all algorithms. Please check and refer these papers:

1. Fang, Z., Wang, Y., Peng, L., & Hong, H. (2021). A comparative study of heterogeneous ensemble-learning techniques for landslide susceptibility mapping. International Journal of Geographical Information Science, 35(2), 321-347.

2. Azarafza, M., Azarafza, M., Akgün, H., Atkinson, P. M., & Derakhshani, R. (2021). Deep learning-based landslide susceptibility mapping. Scientific reports, 11(1), 1-16.

Comment 17: I noticed that the conclusion section tends to repeat abstract and results. The conclusion paragraph should be short, impactful, and direct the reader to this research's next steps and opportunities.

Best regards,

---

## Referee Comment (RC2)

Paper review on "Transfer learning for landslide susceptibility modelling using domain adaptation and case-based reasoning".

This paper evaluates the performance of different transfer algorithms for LSM including case-based reasoning (CBR) and domain adaptation (DA). The study is very interesting, relevant and suitable for GMD. However, the following issues should be carefully addressed before publication:

1. The problem is very well characterized and the objectives clearly established.
2. Authors should explain strategies they have adopted to select non-landslide points from landslide points randomly. What are the criteria and the distance they have set as thresholds for considering non-landslide regions, especially when they have a low-resolution dataset?
3. What is the spatial and temporal resolution of the rainfall dataset? How did the authors handle the spatial resolution difference between the rainfall dataset and DTM-derived parameters? Considering rainfall data's dynamic characteristics, how could authors relate the other statistic parameters (topographic condition, etc.) to dynamic parameters and predict and produce a reliable landslide inventory? How do authors handle the spatio-temporal characteristics of landslide events in the different regions?
   The methodology's major limitation is the different types of mass movements! I want to ask how the author handled and incorporated the geometric differences of different mass movements (landslides) into the models to correctly predict the different types of mass movements, especially knowing that each landslide type has its own geometric and physical characteristics.
4. Finally, Why authors just contended the simple Logistic GAM for implementing the DA algorithm while more robust algorithms exist for solving the non-linearity relationship of the input parameter and also considering the binary case of the classification

---

## Author Comment (AC1)

*Author Responses Addressing Review from Referee #2 for* **"Transfer learning for landslide susceptibility modelling using domain adaptation and case-based reasoning"** *by* **Wang et al.**

For these responses, we address each Referee comment (RC) individually and include our response below it. The Referee Comments are numbered and use a black font, while the Author Responses (AR) use a red font.

This paper evaluates the performance of different transfer algorithms for LSM including case-based reasoning (CBR) and domain adaptation (DA). The study is very interesting, relevant and suitable for GMD. However, the following issues should be carefully addressed before publication:

AR0.0: We want to thank Referee #2 for the helpful comments and suggestions. We have done our best to address each of the comments below.

RC1. The problem is very well characterized and the objectives clearly established.

AR1: Thanks for the comment.

RC2. Authors should explain strategies they have adopted to select non-landslide points from landslide points randomly. What are the criteria and the distance they have set as thresholds for considering non-landslide regions, especially when they have a low-resolution dataset?

AR2: Thanks for the comment.

Following previous work such as Goetz et al. (2011) and Brock et al. (2020), the landslides and non-landslides are selected using simple random sampling and the non-landslide samples are grid cells that did not identify as the landslide. Moreover, because landslides cover a small portion of the entire area, a random sampling of grid cells that do not relate to mapped landslide locations is reasonable to summarize the characteristics of landslide-free zones for the purpose of statistical analyses (Blahut et al., 2010; Goetz et al., 2015; Steger and Glade, 2017).

In our study, one grid cell was selected as one sample point. When the area size of a landslide is less than the area size of a single grid cell, the landslide is removed. We also added the following sentence in *Section 2.3*:

"At the same time, landslides that are smaller than one grid cell were excluded in our study."

Reference:

1. Goetz, J. N., Guthrie, R. H., & Brenning, A. (2011). Integrating physical and empirical landslide susceptibility models using generalized additive models. Geomorphology, 129(3-4), 376-386.

2. Brock, J., Schratz, P., Petschko, H., Muenchow, J., Micu, M., & Brenning, A. (2020). The performance of landslide susceptibility models critically depends on the quality of digital elevation models. Geomatics, Natural Hazards and Risk, 11(1), 1075-1092.

3. Blahut, J., Van Westen, C. J., & Sterlacchini, S. (2010). Analysis of landslide inventories for accurate prediction of debris-flow source areas. Geomorphology, 119(1-2), 36-51.

4. Goetz, J. N., Brenning, A., Petschko, H., & Leopold, P. (2015). Evaluating machine learning and statistical prediction techniques for landslide susceptibility modeling. Computers & geosciences, 81, 1-11.

5. Steger, S., & Glade, T. (2017, May). The challenge of "trivial areas" in statistical landslide susceptibility modelling. In Workshop on World Landslide Forum (pp. 803-808). Springer, Cham.

RC3. What is the spatial and temporal resolution of the rainfall dataset? How did the authors handle the spatial resolution difference between the rainfall dataset and DTM-derived parameters? Considering rainfall data's dynamic characteristics, how could authors relate the other statistic parameters (topographic condition, etc.) to dynamic parameters and predict and produce a reliable landslide inventory? How do authors handle the spatio-temporal characteristics of landslide events in the different regions?

AR3: Thanks for the comment.

We're looking at susceptibility as a static variable and that the potential incorporation of rainfall is a topic for future work that might apply transfer learning beyond the context of landslide susceptibility.

For handle the spatio-temporal characteristics of landslide events in the different regions, Yate et al. (2018) in *TREE* journal pointed out that environmental differences are a key factor for successful model transfer, while spatial and temporal separation may have little effect on model transfer. We considered the environmental characteristics of different regions that have an important influence on the landslide assessment, such as slope, elevation, etc. By comparing the similarity between these characteristics, we can relate landslide events in different regions. Meanwhile, we also delineated in *Discussion*:

"Although the study areas cover a wide range of climates with different land cover types and landslide process types, our set of source areas is by no means complete and the results may therefore not be fully representative for the performances that might be achieved at a global scale. Future work should therefore broaden the database of source areas."

Reference:

1. Yates, K. L., Bouchet, P. J., Caley, M. J., Mengersen, K., Randin, C. F., Parnell, S., ... & Sequeira, A. M. (2018). Outstanding challenges in the transferability of ecological models. Trends in ecology & evolution, 33(10), 790-802.

RC4. The methodology's major limitation is the different types of mass movements! I want to ask how the author handled and incorporated the geometric differences of different mass movements (landslides) into the models to correctly predict the different types of mass movements, especially knowing that each landslide type has its own geometric and physical characteristics.

AR4: Thanks for the comment. This is a really good point. We also discussed it in *Section 4.2*,

"These similarity attributes do not explicitly account for landslide type, which is an important factor to consider when landslide susceptibility modelling (Huang and Zhao, 2018). However, geologic attributes and terrain attributes such as slope angle, may work together as a suitable surrogate to anticipate the most likely landslide types given little to no landslide data in the target area. Landslide type information is also difficult to collect and is often lacking in landslide inventories (Mezaal and Pradhan, 2018). Prior information on unseen areas or integrating expert experience may be helpful in formulating landslide types for transfer learning."

Also, for example, according to the Wieczorek and Jäger (1996) and Zinko et al., 2005, different types of mass movements may depend on the lithology and groundwater and soil moisture conditions in relation to topography. These attributes could also be used for predicting the different types of mass movements. Predicting the landslide type for model transferring is a challenge and still needed to do further research. But we would like to point out that although our study cannot clearly predict landslide types, by identifying landslide types that are common between source and target areas, we can reduce the burden of collecting and labeling data and give a quick landslide susceptibility map that can help decision makers develop basic preventive measures.

Reference:

1. Wieczorek, G. F., & Jäger, S. (1996). Triggering mechanisms and depositional rates of postglacial slope-movement processes in the Yosemite Valley, California. Geomorphology, 15(1), 17-31.

2. Zinko, U., Seibert, J., Dynesius, M., & Nilsson, C. (2005). Plant species numbers predicted by a topography-based groundwater flow index. Ecosystems, 8(4), 430-441.

RC5. Finally, why authors just contended the simple Logistic GAM for implementing the DA algorithm while more robust algorithms exist for solving the non-linearity relationship of the input parameter and also considering the binary case of the classification

AR5: Thanks for the comment.

According to Goetz's and Brock's publications, we can find GAM can obtain good results in terms of predictive performance in landslide assessment compared to several other statistical and machine-learning algorithms. Furthermore, GAM can adjust the degree of non-linearity (or

effective degrees of freedom) for each variable using an inner generalized cross-validation, which can save the time and effort of the calibrating parameters (Wood, 2017). Otherwise, GAM allows for a separate interpretation of additive effects in terms of odds ratios and variable importance, which some existing robust or state-of-the-art algorithms may not be able to do.

But there is research value in considering other state-of-the-art algorithms for implementing the DA.

Reference:

1. Goetz, J. N., Guthrie, R. H., and Brenning, A.: Integrating physical and empirical landslide susceptibility models using generalized additive models, Geomorphology, 129, 376-386, https://doi.org/10.1016/j.geomorph.2011.03.001, 2011.

2. Goetz, J. N., Brenning, A., Petschko, H., and Leopold, P.: Evaluating machine learning and statistical prediction techniques for landslide susceptibility modeling, Computers & Geosciences, 81, 1-11, https://doi.org/10.1016/j.cageo.2015.04.007, 2015.

3. Brock, J., Schratz, P., Petschko, H., Muenchow, J., Micu, M., & Brenning, A. (2020). The performance of landslide susceptibility models critically depends on the quality of digital elevation models. Geomatics, Natural Hazards and Risk, 11(1), 1075-1092.

4. Wood S. 2017. Generalized additive models: an introduction with R. 2nd ed. Chapman and Hall/CRC. London (UK)

---

## Author Comment (AC2)

**Author Responses Addressing Review from Referee #1 for* "Transfer learning for landslide susceptibility modelling using domain adaptation and case-based reasoning" *by* Wang et al.**

For these responses, we address each Referee comment (RC) individually and include our response below it. The Referee Comments are numbered and use a black font, while the Author Responses (AR) use a red font.

Thank you for giving me the opportunity to revise the manuscript entitled "Transfer learning for landslide susceptibility modelling using domain adaptation and case-based reasoning" by Zhihao Wang and his/her colleagues that was submitted to "Geoscientific Model Development". The manuscript used transfer-learning strategies to landslide susceptibility assessment (LAS) for 10 different cases in Austria, Ecuador, and Italy. Application of various learning based algorithms are interesting topics in LSA, but regarding the existed literatures needs to be polished properly.

AR0.0: We want to thank Referee #1 for the helpful comments and suggestions. We have done our best to address each of the comments below.

RC1. The English of the paper is readable; however, I would suggest the authors to have it checked preferably by a native English-speaking person to avoid any mistakes.

AR1: Thanks for the comment and sorry for some mistakes. One of co-authors Jason Goetz is a native speaker, who is from Canada, and we also took another careful look at the revised manuscript.

RC2. Add the "….several case studies" in the title.

AR2: The objective of our study is to evaluate performance of transfer learning in landslide assessments. Hence, we performed experiments to validate the methods in the manuscript based on our available data, rather than specifically targeting these areas.

We might also identify such similarities in the recommended cites below.

For example,

1. Fang, Z., Wang, Y., Peng, L., & Hong, H. (2020). Integration of convolutional neural network and conventional machine learning classifiers for landslide susceptibility mapping. Computers & Geosciences, 139, 104470.

Authors wanted to assess landslide susceptibility by integrating convolutional neural network (CNN) with three conventional machine learning classifiers in the case of Yongxin Country, China. Because they focus on assessing the performance of the method, the case study was used as experimental study area. Therefore, they didn't add case study on the title.

Also, the same is true for the two recommended citations below.

2. Fang, Z., Wang, Y., Peng, L., & Hong, H. (2021). A comparative study of heterogeneous ensemble-learning techniques for landslide susceptibility mapping. International Journal of Geographical Information Science, 35(2), 321-347.

3. Azarafza, M., Azarafza, M., Akgün, H., Atkinson, P. M., & Derakhshani, R. (2021). Deep learning-based landslide susceptibility mapping. Scientific reports, 11(1), 1-16.

Therefore, in order to better reflect the aim of our work, we may not recommend adding "….several case studies" in the title. In Discussion, we also added another paragraph in section 4.3:

"Although the study areas cover a wide range of climates with different land cover types and landslide process types, our set of source areas is by no means complete and the results may therefore not be fully representative for the performances that might be achieved at a global scale. Future work should therefore broaden the database of source areas."

RC3. The concluding remarks of the abstract are not well-written. It's merely the repetition of the objectives and title of the manuscript. in a appropriate abstract, quantitative finding, method limitations, model verification, learning rate and justification have to added. Please kindly modify the abstract.

AR3: Thanks for the comment. According to the preparation of Geoscientific Model Development (GMD) journal, the description for abstract in the Manuscript composition part said:

"The abstract should be intelligible to the general reader without reference to the text. After a brief introduction of the topic, the summary recapitulates the key points of the article and mentions possible directions for prospective research."

For our abstract, the first sentence is to introduce our objective. The rest of it summarizes results and their broader implication, which we believe is appropriate for a Conclusion. The objective of our work is to compare the methods used and analysis their potential for landslide assessments. Hence, based on the GMD description - a concise summary of the key points of the article, we have concisely summarized the experimental results to highlight the purpose of our work.

Therefore, we believe that the abstract in the manuscript can be appropriate for our work.

RC4. Considering the existing literature, the motivation of the manuscript is not convincing. There is a large literature associated with LSA, and various techniques have been applied/developed to assess the susceptibility better. I expect to see a novelty in a manuscript dealing with landslide susceptibility because we already have many case studies. I do not think that new case studies would help the scientific community to go one step further. The result of this manuscript is only valid for the examined area. Based on site-specific conditions and the quality of a landslide inventory, different results can be obtained in another site, and another

technique may appear like the best alternative. However, in the end, these efforts do not help us decide on the best method of LSA. So, please kindly request to provide solid document to support the superiority of the method than other approaches. Please check and refer these papers:

1. Kavzoglu, T., Colkesen, I., & Sahin, E. K. (2019). Machine learning techniques in landslide susceptibility mapping: a survey and a case study. Landslides: Theory, practice and modelling, 283-301.

2. Saleem, N., Huq, M., Twumasi, N. Y. D., Javed, A., & Sajjad, A. (2019). Parameters derived from and/or used with digital elevation models (DEMs) for landslide susceptibility mapping and landslide risk assessment: a review. ISPRS International Journal of Geo-Information, 8(12), 545.

AR4: For our work, we focus on the performance of transfer learning in landslide assessments using a sample of numerous, diverse study areas worldwide. Therefore, it is not a case study.

Transfer learning is the application of knowledge gained from completing one task to help solve a different, but related, problem. In landslide assessments, transfer learning can use the existed landslide inventories to detect landslides in unseen areas. But it should notice that transfer learning techniques have not been systematically explored in the context of landslide susceptibility modeling. Nowadays, more and more research groups try to scale modeling effort up from the local scale to the regional or even global scale, e.g.:

1. Lima, P., Steger, S. & Glade, T. Counteracting flawed landslide data in statistically based landslide susceptibility modelling for very large areas: a national-scale assessment for Austria. Landslides 18, 3531–3546 (2021).

2. Lin, Q., Lima, P., Steger, S., Glade, T., Jiang, T., Zhang, J., ... & Wang, Y.. National-scale data-driven rainfall induced landslide susceptibility mapping for China by accounting for incomplete landslide data. Geoscience Frontiers, 12(6), 101248 (2021).

3. Martina Wilde, Andreas Günther, Paola Reichenbach, Jean-Philippe Malet & Javier Hervás Pan-European landslide susceptibility mapping: ELSUS Version 2, Journal of Maps, 14:2, 97-104 (2018).

4. Lin, L., Lin, Q., and Wang, Y.: Landslide susceptibility mapping on a global scale using the method of logistic regression, Nat. Hazards Earth Syst. Sci., 17, 1411–1424 (2017).

Hence, it is of great importance to research scale modeling in landslide assessments. Our study is to study the landslide modeling in global scale using transfer learning, which can provide some meaningful information for the researchers who are interested in this field.

We cited Kavzoglu paper in the Introduction:

"Machine learning is currently the most commonly applied method in research for solving the problem of landslide prediction (Goetz et al., 2015; Merghadi et al., 2020; Kavzoglu et al., 2019)."

RC5. The necessity & novelty of the manuscript should be presented and stressed in the "Introduction" section. Please check and refer these papers:

1. Nanehkaran, Y. A., Mao, Y., Azarafza, M., Kockar, M. K., & Zhu, H. H. (2021). Fuzzy-based multiple decision method for landslide susceptibility and hazard assessment: A case study of Tabriz, Iran. Geomechanics and Engineering, 24(5), 407-418.

2. Ngo, P. T. T., Panahi, M., Khosravi, K., Ghorbanzadeh, O., Kariminejad, N., Cerda, A., & Lee, S. (2021). Evaluation of deep learning algorithms for national scale landslide susceptibility mapping of Iran. Geoscience Frontiers, 12(2), 505-519.

AR5: Thanks for providing the papers to refer.

In the Introduction section, we cited Lin et al. (2021) to clarify the difficulty of establishing the landslide inventory data for model training and testing. Then, we used Sequeira et al. (2016), Wenger and Olden (2012), and Rudy et al. (2016) to verify the model transferability have the potential to help solve the above problem in landslide assessments. According to Yates et al. (2018) study, direct transfer of traditional machine learning models is problematic in landslide studies. A successful model transfer does not necessarily rely solely on the extent of geographic or temporal separation, but rather on the similarity of the environmental conditions between the source and target areas. Meanwhile, following Shimodaira (2000), Pan and Yang (2010), and Yates et al. (2018), model performance can be degraded due to differences in feature space and/or data distributions. Hence, our study aims at solving these problems by using case-based reason and domain adaptation. These references could underline the necessity and novelty of this work in the Intro section.

Because Nanehkaran et al. (2021) and Ngo et al. (2021) studies are not related to transfer learning, we might not cite them in our manuscript. But these great works could be useful for our further study.

References:

1. Lin, Q. G., Lima, P., Steger, S., Glade, T., Jiang, T., Zhang, J. H., Liu, T. X., and Wang, Y.: National-scale data-driven rainfall induced landslide susceptibility mapping for China by accounting for incomplete landslide data, Geoscience Frontiers, 12, 101248, https://doi.org/10.1016/j.gsf.2021.101248, 2021.

2. Sequeira, A. M. M., Mellin, C., Lozano-Montes, H. M., Vanderklift, M. A., Babcock, R. C., Haywood, M. D. E., Meeuwig, J. J., and Caley, M. J.: Transferability of predictive models of coral reef fish species richness, Journal of Applied Ecology, 53, 64-72, https://doi.org/10.1111/1365-2664.12578, 2016.

3. Wenger, S. J. and Olden, J. D.: Assessing transferability of ecological models: an underappreciated aspect of statistical validation, Methods in Ecology and Evolution, 3, 260-267, https://doi.org/10.1111/j.2041-210X.2011.00170.x, 2012.

4. Rudy, A. C. A., Lamoureux, S. F., Treitz, P., and van Ewijk, K. Y.: Transferability of regional permafrost disturbance susceptibility modelling using generalized linear and generalized additive models, Geomorphology, 264, 95-108, https://doi.org/10.1016/j.geomorph.2016.04.011, 2016.

5. Shimodaira, H.: Improving predictive inference under covariate shift by weighting the log-likelihood function, J Stat Plan Infer, 90, 227-244, https://doi.org/10.1016/S0378-3758(00)00115-4, 2000.

6. Pan, S. J. and Yang, Q. A.: A Survey on Transfer Learning, IEEE Transactions on knowledge and data engineering, 22, 1345-1359, https://doi.org/10.1109/TKDE.2009.191, 2010.

7. Yates, K. L., Bouchet, P. J., Caley, M. J., Mengersen, K., Randin, C. F., Parnell, S., Fielding, A. H., Bamford, A. J., Ban, S., Barbosa, A., Dormann, C. F., Elith, J., Embling, C. B., Ervin, G. N., Fisher, R., Gould, S., Graf, R. F., Gregr, E. J., Halpin, P. N., Heikkinen, R. K., Heinanen, S., Jones, A. R., Krishnakumar, P. K., Lauria, V., Lozano-Montes, H., Mannocci, L., Mellin, C., Mesgaran, M. B., Moreno-Amat, E., Mormede, S., Novaczek, E., Oppel, S., Crespo, G. O., Peterson, A. T., Rapacciuolo, G., Roberts, J. J., Ross, R. E., Scales, K. L., Schoeman, D., Snelgrove, P., Sundblad, G., Thuiller, W., Torres, L. G., Verbruggen, H., Wang, L., Wenger, S., Whittingham, M. J., Zharikov, Y., Zurell, D., and Sequeira, A. M. M.: Outstanding Challenges in the Transferability of Ecological Models, Trends in ecology & evolution, 33, 790-802, https://doi.org/10.1016/j.tree.2018.08.001, 2018.

RC6. Provide a literature of the methods used learning-based LSA methods in "Introduction". The use of a table to demonstrate the advantage disadvantage of these methods can be useful. Towards the end, mention the superiority & repeat the novelty of your work. Please check and refer these papers:

1. Wang, Y., Fang, Z., & Hong, H. (2019). Comparison of convolutional neural networks for landslide susceptibility mapping in Yanshan County, China. Science of the total environment, 666, 975-993.

2. Fang, Z., Wang, Y., Peng, L., & Hong, H. (2020). Integration of convolutional neural network and conventional machine learning classifiers for landslide susceptibility mapping. Computers & Geosciences, 139, 104470.

AR6: Because comparison of modeling techniques is not our objective, dedicating too much space to the presentation of various machine-learning techniques would distract the reader from the main topic of this paper. We used several citations to clarify the popularity of machine learning in landslide assessments and some papers have researched performance of different

methods in landslide assessments, e.g., Goetz et. al (2015) in Computers & Geosciences. Hence, readers could find more detailed in those specific papers which are to demonstrate the advantage and disadvantage of learning-based LSA methods.

Reference:

1. Goetz, J. N., Brenning, A., Petschko, H., and Leopold, P.: Evaluating machine learning and statistical prediction techniques for landslide susceptibility modeling, Computers & Geosciences, 81, 1-11, https://doi.org/10.1016/j.cageo.2015.04.007, 2015.

RC7. A relevant source of subjectivity and uncertainty is introduced when splitting the input parameters into an arbitrary number of classes with random break values. These choices affect the results. Would you please describe your solution?

AR7: Thanks for the question. We're not splitting the predictor variables in this study and our results are therefore not affected by such decisions. Instead, we use the GAM, which optimizes the complexity of its transformation functions in a data-driven way using the training data.

RC8. The methodology section is weakly written. So, my suggestion is to reconstruct it. In addition, please kindly use the flowcharts, model verification, benchmarks and error tables to provide detailed methodology.

AR8: Thanks for this comment. We insert a short paragraph to section 2, before subsection 2.1 to further clarify the methodology used in our study:

"In this section we first introduce the general CBR and DA methods separately (sections 2.1 and 2.2). We then explain how CBR and DA models as well as the combined CBR-DA approach are trained and tested in this work (section 2.3). The data used for demonstrating the proposed approaches is then briefly presented, referring the readers to the relevant local literature for details (section 2.4)."

Meanwhile, the flowchart (Figure 1) listed the completed and detailed processes of all methods used in our study. Hence, with the additional statements and flowchart, methodology shows the details better.

RC9. What will be happened if you use this algorithm for another region?

AR9: In our manuscript, we use other regions similar to the target region to obtain a landslide model, and then use this model to make predictions for the target region. We cannot predict with certainty when our algorithm is applied to other regions, but our assessment strategy shows the range of performances we may obtain using a wide range of geographically diverse source and target regions.

For example, in our manuscript, the AUROCs of a model trained by the most related source area Piacenza 25 was 0.762 and that of the model trained with Bologna 25 data itself was also

0.762. Otherwise, on average AUROCs obtained by using our algorithm were below the overoptimistic situation (training and testing in target area) by only ~0.05.

Hence, we might be able to obtain the good result when we use this algorithm for another region.

RC10. Please add a subsection clearly articulating the main limitations, wider applicability of your methods, and findings in the "Discussion" section. Also, the authors should deepen the discussion. Please check and refer these papers:

1. Wu, Y., Ke, Y., Chen, Z., Liang, S., Zhao, H., & Hong, H. (2020). Application of alternating decision tree with AdaBoost and bagging ensembles for landslide susceptibility mapping. Catena, 187, 104396.

2. Azarafza, M., Ghazifard, A., Akgün, H., & Asghari-Kaljahi, E. (2018). Landslide susceptibility assessment of South Pars Special Zone, southwest Iran. Environmental Earth Sciences, 77(24), 1-29.

AR10: Thanks for the comment. We changed the title of section 4.3 to "Utility of domain adaptation in geospatial learning and other limitations". In Discussion, we pointed out limitations for our methods (e.g., "DA did not generally improve transfer learning performance"). It might be clear to describe the limitation for each method used in our paper. For general limitations for all methods used in our study, we added another paragraph in section 4.3:

"Although the study areas cover a wide range of climates with different land cover types and landslide process types, our set of source areas is by no means complete and the results may therefore not be fully representative for the performances that might be achieved at a global scale. Future work should therefore broaden the database of source areas."

We also thank the reviewer for pointing out the potential of other ML techniques for landslide susceptibility modelling by providing the papers. There are some papers to assess and compare different ML techniques in landslide assessments, for example, Goetz et. al (2015) in Computers & Geosciences showed that there's usually not much of a performance difference.

RC11. Please define susceptibility as this terminology is poorly understood by many researchers.

AR11: Thanks for the comment. We added the sentences in Intro section:

"Landslide susceptibility refers to the likelihood of landslides occurring in the region, depending on the local topographic conditions, and estimating the likely location of landslides (Reichenbach et al., 2018)."

RC12. There are no LSA maps that indicate the degree of the susceptibility, please add the required map from triggering factors to risk map. In LSA analysis, providing maps is very important to understand the authors' results.

AR12: The figure 9 showed an example of classified landslide susceptibility maps. Meanwhile, the objective of this study was to evaluate a methodology, not to create susceptibility maps for areas in which these maps already exist from previous studies. Also note that this would require us to include a huge number of maps for a total of 17 target areas and for different modelling strategies, which would not be possible even in supplementary material.

RC13. Please provide justifications for your model, I believe application of other learning based method can be providing much better results (for example: deep learning). Please verify your model with well-known methods.

AR13: Thanks for the suggestion. According to Goetz's publications, we can find GAM also can obtain good results in terms of predictive performance.

Reference:

1. Goetz, J. N., Guthrie, R. H., and Brenning, A.: Integrating physical and empirical landslide susceptibility models using generalized additive models, Geomorphology, 129, 376-386, https://doi.org/10.1016/j.geomorph.2011.03.001, 2011.

2. Goetz, J. N., Brenning, A., Petschko, H., and Leopold, P.: Evaluating machine learning and statistical prediction techniques for landslide susceptibility modeling, Computers & Geosciences, 81, 1-11, https://doi.org/10.1016/j.cageo.2015.04.007, 2015.

RC14. What type landslide was analyzed? Needs to describe.

AR14: Please find the relevant information in table 3 and we are sorry for not providing more details here, because the cited local literature for each source area could provide more information.

RC15. Please add the "case study" section and provide relevant information about the geology, morphology, seismicity, hydrology and other triggering factors that impact on LSA.

AR15: Please find the section 2.4 Case study transfer source and target areas and table 3. We apologize for not being able to present detailed information in this kind of publication, which can be found in the cited local literature for each source area.

RC16. Please provide the table of hyper-parameters values of all algorithms. Please check and refer these papers:

1. Fang, Z., Wang, Y., Peng, L., & Hong, H. (2021). A comparative study of heterogeneous ensemble-learning techniques for landslide susceptibility mapping. International Journal of Geographical Information Science, 35(2), 321-347.

2. Azarafza, M., Azarafza, M., Akgün, H., Atkinson, P. M., & Derakhshani, R. (2021). Deep learning-based landslide susceptibility mapping. Scientific reports, 11(1), 1-16.

AR16: Thanks for providing the papers.

A generalized additive model is a generalized linear mode with a linear predictor involving a sum of smooth functions of covariates. According to the Wood's book, GAM has no hyperparameters since the complexity of transformation functions is optimized internally (on the training set) in a data-driven way.

Reference:

1. Wood, S. N. (2006). Generalized additive models: an introduction with R. chapman and hall/CRC.

RC17. I noticed that the conclusion section tends to repeat abstract and results. The conclusion paragraph should be short, impactful, and direct the reader to this research's next steps and opportunities.

AR17: Thanks for the comment.

For our abstract, the first sentence is to introduce our objective. The rest of it summarizes results and their broader implication, which we believe is appropriate for a Conclusion. Therefore, we describe more details in Conclusion to show reader more clearly about our study's aim, results, and further applications.

---

## Author Response (AR3)

*Author Responses Addressing Review from Editor for* "Transfer learning for landslide susceptibility modelling using domain adaptation and case-based reasoning" *by* **Wang et al.**

We would like to thank the editor and the reviewers for their thorough revisions and the thoughtful comments they provided. We have made extensive changes in order to incorporate recent publications related to transfer learning and deep learning in both the introduction and the discussion, and an English native speaker carefully revised and improved the language. We therefore hope that the editor will be satisfied with the changes made in the revised manuscript.

In our responses, we address each Editor comment individually and include our response below it. The Editor Comments (EC) are numbered and use a black font, while the Author Responses (AR) use a red font.

Dear authors:

Thank you for participating in the Interactive Discussion. You revised the paper to make it more consistent with the arguments of the referee. As GMD's editor, you'll notice that one referee rated the scientific significance as "Fair" and the scientific quality as "Poor." Numerous technical aspects have been raised and must be thoroughly addressed. Several additional comments follow, which I recommend you consider before publishing.

AR0.0: We want to thank Editor for the comments. We have done our best to address each of the comments below.

**EC1:** This paper's motivation is unconvincing. Except for comparing and combining two transfer learning strategies for landslide susceptibility modeling, case-based reasoning (CBR) and domain adaptation (DA), the authors have not provided any new methods or made substantive improvements to the existing models. One of the innovations could result from evaluating the potential of transfer learning for landslide susceptibility modeling using CBR and DA techniques to reduce the burden of data collection and labeling. However, as far as I am aware, the following key related works have been proposed in recent years (but are not limited to):

1. Wang, H., Wang, L., and Zhang, L.: Transfer learning improves landslide susceptibility assessment, Gondwana Research, 1-17, https://doi.org/10.1016/j.gr.2022.07.008, 2022.
2. Xu, Q., Ouyang, C., Jiang, T., Yuan, X., Fan, X., and Cheng, D.: MFFENet and ADANet: a robust deep transfer learning method and its application in high precision and fast cross-scene recognition of earthquake-induced landslides, Landslides, 19(7), 1617-1647, 2022.
3. Ai, X., Sun, B., and Chen, X.: Construction of small sample seismic landslide susceptibility evaluation model based on Transfer Learning: a case study of Jiuzhaigou earthquake, Bulletin of Engineering Geology and the Environment, 81(3), 116, https://doi.org/10.1007/s10064-022-02601-6, 2022.
4. Qin, S., Guo, X., Sun, J., Qiao, S., Zhang, L., Yao, J., Cheng, Q., and Zhang, Y.: Landslide detection from open satellite imagery using distant domain transfer learning, Remote Sensing, 13(17), 3383, https://doi.org/10.3390/rs13173383, 2021.
5. Liu, D., Li, J., and Fan, F.: Classification of landslides on the southeastern Tibet Plateau based on transfer learning and limited labelled datasets, Remote Sensing Letters, 12(3), 286-295, 2021.
6. Zhu, Q., Chen, L., Hu, H., Pirasteh, S., Li, H., and Xie, X.: Unsupervised Feature Learning to Improve Transferability of Landslide Susceptibility Representations, IEEE Journal of Selected Topics in Applied Earth Observations and Remote Sensing, 13, 3917-3930, 2020.

7. Lu, H., Ma, L., Fu, X., Liu, C., Wang, Z., Tang, M. and Li, N.: Landslides information extraction using object-oriented image analysis paradigm based on deep learning and transfer learning, Remote Sensing, 12(5), 752, https://doi.org/10.3390/rs12050752, 2020.

The author should give a broad discussion on these transfer learning methods applied to landslide susceptibility modelling. And then, the necessity and novelty of this work should be demonstrated by analyzing and summarizing the advantages and disadvantages of these transfer learning methods.

**AR1**: Thanks for the comments and providing the references. It is helpful for us to clarify the research value of our study. Our response is divided into two parts, the first part is the response to the comment (Part 1) and the second part is the phrases we added to the manuscript (Part 2).

Part 1:

We would like to kindly point out that the long list of recent transfer learning publications also shows that this is a very active research topic, that is, transfer learning techniques have not been fully explored in landslide susceptibility modelling.

In the field of landslide assessment research, environmental characteristics are extremely important for landslide modeling, prediction of landslide susceptibility in unknown areas, and interpretation of final results. As Yates et al. (2018) point out in their article:

"Environmental dissimilarity is what matters most for successful transfers, for which spatio-temporal distances might only occasionally be good surrogates."

However, when we read the articles above and other related papers, we can easily find that they considered target and source areas with similar environmental characteristics/data distribution, which are inconsistent with realistic landslide assessments. For example, Wang et al. conducted a transfer learning study only for the region of Hong Kong; Xu et al. specified the landslide triggered by earthquakes; Ai et al. implemented landslide model transfer only for Sichuan Province; Liu et al. studied only the south-eastern Tibet Plateau; Zhu et al. studied the Chongqing region; Lu et al. studied only the region located in Sichuan. In contrast, our work explores the use of model transfer using data from different parts of the globe.

Hence, it is of great importance to research applying model transfer with data from different regions and data sources around the globe, which is one of our study motivations and innovations. Also, we have further pointed this out in the Discussion.

"Until now, model transfer in landslide modelling have usually relied on a homogeneous availability of data and a strong model generalization to avoid local overfitting and allow the application of a model in an adjacent target region (Goetz et al., 2011; Wenger and Olden, 2012; Petschko et al., 2014; Bordoni et al., 2020). Although this approach has been identified as a robust method for regional susceptibility modelling, its model transferability is often limited to nearby locations that have the same feature space and a nearly identical data distribution. "

Meanwhile, in general, data spatial resolution affects model fit and prediction. Previous work using transfer learning for landslide assessments only uses the same spatial resolution data for achieving landslide model transfer. However, as our work demonstrates, we do not necessarily need to be bound to data with similar spatial resolution. For example, we found that even though a source area had a different resolution than the target area, the model transfer performance was still great (target area: Burgenland in Austria with a 10 m resolution, source area: Modena in

Italy with a 25 m resolution). Thus, evaluating different spatial resolution for landslide model transfer is also one of our study motivations and innovations. We also pointed it out in the manuscript:

"We evaluate the performance of transferred susceptibility models using DA, CBR and a combined CBR-DA technique, as well as the sensitivity of these methods to spatial resolution."

Otherwise, most studies have been conducted based on a single source area, while our paper discusses the case of single and multiple source areas and proposes the use of similarity between source and target areas as weight values to combine landslide models obtained from multiple source areas. Thus, the experiment and the obtained results further improve the comprehension of landslide model transfer studies.

Part 2:

We have extended the Introduction to demonstrate recent transfer learning methods (Line 47):

[revised manuscript text omitted]

**EC2:** More detailed information, such as novelty, valuable results, and the major limitation of the methodology, should be included in the conclusion. The authors should also suggest several future research directions.

**AR2**: Thanks for the comment. The Conclusion has been reconstructed to provide more detailed information based on the editor's comments (Line 446):

"The aim of our study was to examine the performances of geographically informed case-based reasoning (CBR) and unsupervised domain adaptation (DA) in geographically transferring knowledge for landslide susceptibility modelling in "new" target areas without landslide inventory data. We extended the study of landslide model transfers to a larger global scale and considered the effect of different spatial resolutions on landslide model transfer. In addition, different scenarios (single source area and multiple source areas) were considered, which made methods and results much closer to practical applications in the real world. Moreover, in the multi-source scenario, we proposed a method to combine multiple landslide models based on environmental similarity. Our comparative study revealed that CBR strategies with a single source area and multiple related source areas were robust and effective in developing highly transferable landslide susceptibility models without requiring prior knowledge of landslides in the target area. In particular, single-source CBR was the most effective method for performing model transfer to the target area in most situations. Its performance was also very close to that obtained by models trained with data from the target area itself. CBR similarity criteria in our study are still preliminary, and data sets used in our study might not be enough for an application at a global scale, which should therefore be considered in future research.

Overall, the findings of this paper demonstrated that the proposed transfer leaning approaches can alleviate the burden of collecting and labelling data, resulting in a more expedited preparation of landslide susceptibility maps for large and data-scarce regions. By calculating the similarity between data and region characteristics, trained models can directly be used for the new task, especially in situations that require rapid model development, such as emergency situations. We also suggest that novel methods such as deep learning may also benefit greatly for landslide model transfer studies."

**EC3:** Throughout the paper, English writing should be greatly improved. Some sentences are too complex, and the correct meaning cannot be extracted. Line 25: "Landslide susceptibility refers to..., and to estimating the likely location of future landslides," for example, but there are many more. I strongly advise the authors to to go over and revise their manuscript.

**AR3**: Thanks for the comment. We have tried our best to correct and improve our English writing. We also had the paper reviewed and corrected carefully by a (Canadian) English native speaker. There are many revisions (more than 200) in the revised manuscript, and here are some examples of the revisions. All line numbers were based on the revised manuscript with tracking.

1. We changed Line 24: "Landslide susceptibility refers to..., and to estimating the likely location of future landslides," to Line 27 "These models are typically data-driven and rely heavily on terrain characteristics to capture conditions that can lead to landslide occurrence.".

2. We reconstructed Line 57 "Transfer learning techniques such as domain adaptation (DA) and case-based reasoning (CBR) have been developed to select the best data and corresponding models from source areas for predicting in a spatially and or temporally distinct target area." to
   "Transfer learning techniques such as domain adaptation (DA) and case-based reasoning (CBR) are emerging techniques to tackle the challenge of model transfer. In general, they have been developed to select the most suitable data and corresponding models from source areas with similar data characteristics for predicting to a distinct target area in space and time."

3. Line 122 "the problem part for formalizing" to "the challenge of formalizing"

4. Line 149 "A latent feature space is defined in which the source and target areas have the same distribution, and as a consequence, classifiers trained on labelled data from source areas are likely to perform well in the target area" to "At first, a latent feature space is defined in which the source and target areas have the same distribution;, and as a consequence, classifiers trained on labelled data from source areas are likely to perform well in the corresponding target area".

5. Line 287 "The distribution trend it displayed implied that single-source DA to some extent improved performances" to "This distribution trend implied to some extent that single-source DA improved performances".